# One-step synthesis of single-site vanadium substitution in 1T-WS$_2$ monolayers for enhanced hydrogen evolution catalysis

Ali Han[1,2,9], Xiaofeng Zhou[1,3,9], Xijun Wang [4,9], Sheng Liu[5], Qihua Xiong [5,6], Qinghua Zhang[7], Lin Gu[7], Zechao Zhuang[2], Wenjing Zhang[8], Fanxing Li [4], Dingsheng Wang [2✉], Lain-Jong Li [1✉] & Yadong Li[2]

Metallic tungsten disulfide (WS$_2$) monolayers have been demonstrated as promising electrocatalysts for hydrogen evolution reaction (HER) induced by the high intrinsic conductivity, however, the key challenges to maximize the catalytic activity are achieving the metallic WS$_2$ with high concentration and increasing the density of the active sites. In this work, single-atom-V catalysts (V SACs) substitutions in 1T-WS$_2$ monolayers (91% phase purity) are fabricated to significantly enhance the HER performance via a one-step chemical vapor deposition strategy. Atomic-resolution scanning transmission electron microscopy (STEM) imaging together with Raman spectroscopy confirm the atomic dispersion of V species on the 1T-WS$_2$ monolayers instead of energetically favorable 2H-WS$_2$ monolayers. The growth mechanism of V SACs@1T-WS$_2$ monolayers is experimentally and theoretically demonstrated. Density functional theory (DFT) calculations demonstrate that the activated V-atom sites play vital important role in enhancing the HER activity. In this work, it opens a novel path to directly synthesize atomically dispersed single-metal catalysts on metastable materials as efficient and robust electrocatalysts.

[1] Physical Sciences and Engineering Divison, King Abdullah University of Science and Technology, Thuwal 23955-6900, Kingdom of Saudi Arabia. [2] Department of Chemistry, Tsinghua University, Beijing 100084, China. [3] Shenzhen Chang Long Technology Co., Ltd, Shenzhen 518117, China. [4] Department of Chemical and Biomolecular Engineering, North Carolina State University, Raleigh, NC 27606, USA. [5] Division of Physics and Applied Physics, School of Physical and Mathematical Sciences, Nanyang Technological University, Nanyang Ave, Singapore 637371, Singapore. [6] State Key Laboratory of Low-Dimensional Quantum Physics and Department of Physics, Tsinghua University, Beijing 100084, China. [7] Beijing National Laboratory for Condensed Matter Physics, Institute of Physics, Chinese Academy of Sciences, Beijing 100190, China. [8] SZU-NUS Collaborative Innovation Center for Optoelectronic Science & Technology, Key Laboratory of Optoelectronic Devices and Systems of Ministry of Education and Guangdong Province, College of Optoelectronic Engineering, Shenzhen University, Shenzhen 518060, China. [9] These authors contributed equally: Ali Han, Xiaofeng Zhou, Xijun Wang. ✉email: wangdingsheng@mail.tsinghua.edu.cn; lance.li@kaust.edu.sa

Hydrogen fuel generation from water splitting is one of the most promising ways to replace conventional fossil fuels and solve the energy crisis[1,2]. Recently, various strategies have been developed to realize the highly efficient catalysts for hydrogen evolution reaction (HER), including semiconductor-based photocatalytic HER (polymer g-$C_3N_4$[3], Ag/semiconductor[4], etc.), photoelectrochemically catalytic HER[5], and metal-based electro-catalytic HER (metal sulfides[1,6,7], metal carbides[8,9], etc.). Commercially, noble metals from the Pt group are utilized to reduce the overpotential of HER and boost the kinetics with unrivaled activities, however, they usually suffer from scarcity, high-cost, and long-term instability. Hence, it is highly desirable to explore robust and efficient HER alternative catalysts with earth-abundant elements to realize the hydrogen economy.

Transition metal dichalcogenides (TMDs) from Group VI elements have recently kindled tremendous investigation as efficient Pt substitutes for HER catalysis because of the catalytically active S atoms on edge sites[10]. Unfortunately, the high proportion of inactive basal plane of $MX_2$ (M = Mo or W, X = S or Se) significantly limits the catalytic performance because of the low electronic transfer capability, leading to the sluggish electrocatalytic kinetics[1]. Two key factors are worthy of being considered to maximize HER activity. One is to increase the metallic phase proportion of $MX_2$, thus improving the intrinsic conductivity of $MX_2$ and boosting HER activity[2,6,11]. However, it is still challenging to directly synthesize a highly pure metallic phase, especially for $1T-WS_2$, owing to the highest formation energy of $1T-WS_2$ (0.89 eV per formula) among all the polymorphs of $MX_2$ (Supplementary Fig. 1)[12]. Recently, the phase-engineered syntheses of metallic $MX_2$ from the 2H phase have been widely developed via wet-chemistry or exfoliated methods, however, the as-produced metallic $MX_2$ domains were usually found with 1T(1T′)/2H mixed-phase and unstable after long-time air-exposure[1,2,6,7,13–22]. The other important factor is increasing the density of active sites of $MX_2$. The field of single atomic catalysts (SACs) comprising isolated metal atoms on the varied supports gives new opportunities for the development of $MX_2$ with increased active sites due to the high atom utilization of SACs[23–25]. However, to the best of

our knowledge, the reports on the direct one-step vapor-phase synthesis of SACs on the highly pure metallic $MX_2$ have not been explored.

Here, we show highly dispersed single vanadium atoms on the $1T-WS_2$ monolayers (denoted as V SACs@1T-$WS_2$), which are synthesized through a one-step chemical vapor deposition (CVD) via controlling the introduction of $VCl_3$. By using $VCl_3$ as the co-precursor, the metallic tungsten disulfide ($WS_2$) monolayers show an ultrahigh 1T concentration of 91%, which is the highest 1T ratio achieved by CVD so far (Supplementary Table 1). Remarkably, the V SACs@1T-$WS_2$ monolayers show superior HER activity comparable to their 2H counterparts, with a low Tafel slope of 61 mV/dec and high turnover frequency (TOF) of 3.01 $s^{-1}$ at 100 mV, and a remarkable long-term catalytic stability. We also demonstrate that the highly active single-atom V sites play a vital role in enhancing the HER activity of intrinsic $1T-WS_2$ monolayers.

## Results

**Characterization of V SACs@1T-$WS_2$ monolayers.** Figure 1 shows the scheme for the growth of V SACs@1T-$WS_2$ monolayers using $WO_3$, sulphur, and $VCl_3$ as the co-precursors. For comparison, 2H-$WS_2$ monolayers were prepared without using the $VCl_3$ co-precursor at the same condition and the growth process has been reported elsewhere[26–33]. Strikingly, a black film of $V_2O_3$ (Fig. 2a and Supplementary Fig. 2) was observed on the sapphire surface during the V SACs@1T-$WS_2$ monolayers growth, while no $V_2O_3$ film appeared for the 2H-$WS_2$ monolayers growth. The optical micrographs of fresh sapphire and $V_2O_3$ film were also provided here for comparison, as seen in Supplementary Fig. 3a, b. In sharp contrast to the triangular morphology of 2H-$WS_2$ (Fig. 2b), the V SACs@1T-$WS_2$ monolayers show a uniformly circular morphology with a lateral size of 30 μm (Fig. 2b and Supplementary Fig. 3c).

The atomic structure of V SACs@1T-$WS_2$ monolayers was investigated by the aberration-corrected STEM, as provided in Fig. 2c, d, showing that the evident Z-contrast intensity sites are

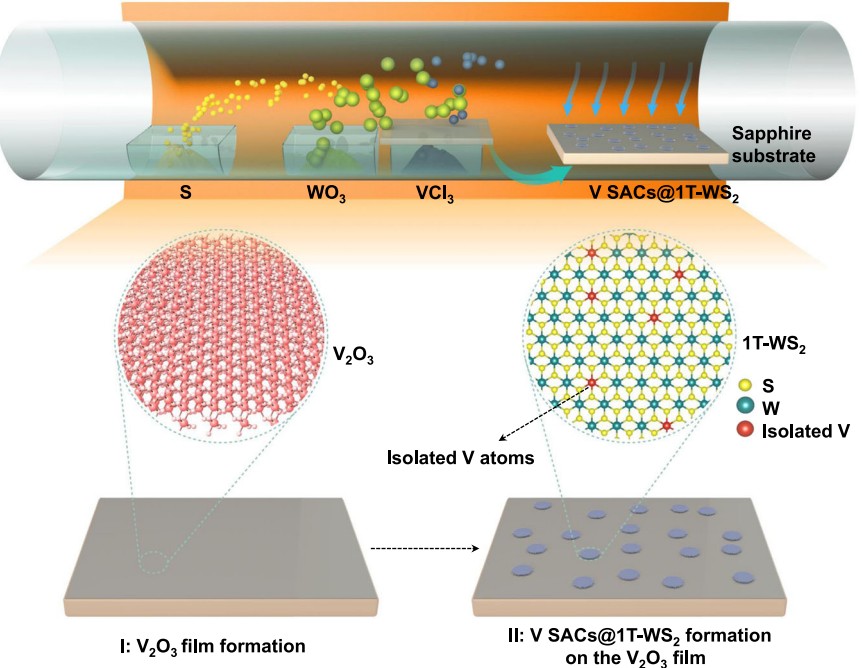

**Fig. 1 Schematic illustration of the synthetic process for V SACs@1T-$WS_2$ monolayers. I** $V_2O_3$ formation in the early growth stage. **II** V SACs@1T-$WS_2$ monolayers formed on the $V_2O_3$ film.

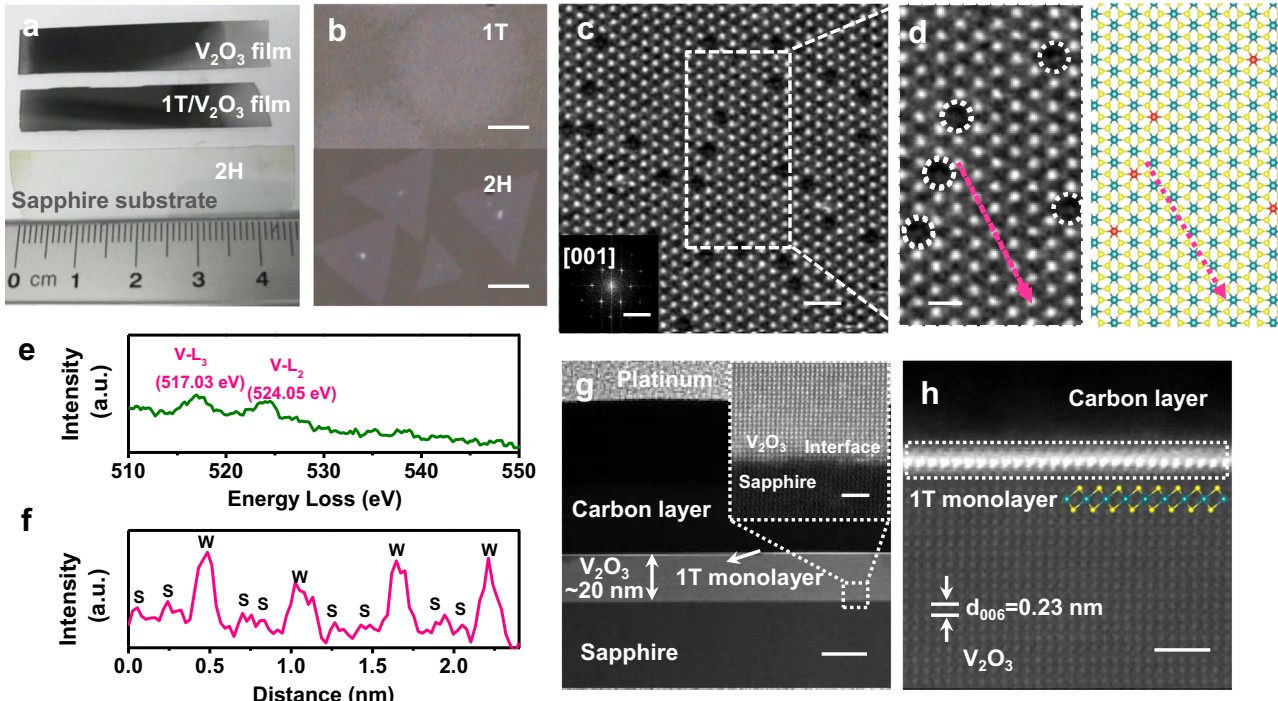

**Fig. 2 Characterization of V SACs@1T-WS₂ monolayers. a** Pictures of different materials formed on the sapphire substrates. Up: V₂O₃ film; Middle: V SACs@1T-WS₂/V₂O₃ film; down: 2H-WS₂. **b** Optical micrographs of as-grown V SACs@1T-WS₂ (up) and 2H-WS₂ (down) grown by CVD. **c** High-resolution HAADF-STEM image for the V SACs@1T-WS₂. Inset: the corresponding fast Fourier transform (FFT) of (**c**). **d** Left: zoom-in high-resolution STEM image for the V SACs@1T-WS₂ indicated by (**c**). Right: the schematic model of V SACs@1T-WS₂. The blue spheres, yellow spheres, and red spheres represent the W atoms, S atoms, and V atoms, respectively. **e** EELS spectrum of vanadium element in the V SACs@1T-WS₂ monolayer. **f** Intensity profiles of the pink dashed arrow indicated by Fig. 1d. **g** Low-resolution cross-section HAADF-STEM image of V SACs@1T-WS₂/V₂O₃ film on the sapphire substrate. Inset: the interface between V₂O₃ and sapphire substrate. **h** High-resolution HAADF-STEM image of V SACs@1T-WS₂ on the V₂O₃ film; scale bars: (**b**) up: 10 μm, down: 5 μm; (**c**) 1 nm, inset: 5 1/nm; (**d**) 0.5 nm; (**g**) 20 nm, inset: 1 nm; (**h**) 1 nm.

strongly dependent on the atomic weight. Figure 2d shows a magnified image of the region outlined by a white dashed rectangle in Fig. 2c. Both the STEM images display a hexagonal packing along [001] zone axis which is usually observed in monolayered 2H-WS₂ (Supplementary Fig. 4a), as highlighted by the fast Fourier transform (FFT, Fig. 2c inset), implying no 1T′ reconstruction occurred[1,18]. In particular, only W atoms are identified because of a much lower atomic number of S compared with W, and the invisible S atoms were also illustrated in the reported metallic WS₂ crystals[1,16,22]. In addition to the invisible S atoms, the atomic positions where W atoms are replaced by V atoms can also be obscurely seen due to the significantly reduced contrast at the W atomic sites (marked by the white dashed circles). The electron energy-loss spectroscopy (EELS) of V SACs@1T-WS₂ in Fig. 2e shows two major features of V-L₂ (524.05 eV) and V-L₃ (517.03 eV) peaks assigned to V⁴⁺, affirming the V substitutions in the 1T-WS₂ layer[34]. The EELS of S and O spectra in Supplementary Fig. 5 were shown to further reveal the presence of S and absence of O in the transferred 1T samples, excluding the V signals from V-based oxidations. The schematic atomic structure of V SACs@1T-WS₂ is depicted in Fig. 2d (right), in which we replace W atoms (blue spheres) of WS₂ with V atoms (red spheres) to illustrate isolated V atoms in 1T-WS₂. Moreover, the W–S–S intensity sequence in Fig. 2f (from the pink dashed arrow in Fig. 2d) is also indicative of the 1T phase of WS₂ due to a misaligned top and bottom S atom. In contrast, for the 2H-WS₂ in Supplementary Fig. 4b, the S atoms at the top and bottom sublayers are overlapped, leading to an alternating W and S intensity pattern. In addition,

W–S–S–V–S–S–W and W–V–W intensity profile sequences from both experimental and simulated STEM images are also achieved to verify the V atoms replacement at W sites (see details in Supplementary Figs. 6 and 7). It is anticipated that the V and S atoms could become visible if the 1T phase is transformed into the 2H phase, which has been shown in a pre-published literature[35]. To further confirm the V-atom concentration in the 1T-WS₂ layer, we anneal the 1T-WS₂ sample at 200 °C (in the air for 30 min) for STEM imaging and find that the 1T phase is completely transformed into the 2H phase, as shown in Supplementary Figs. 8 and 9. Simultaneously, both the V atoms and S atoms are prominently discernible, in consistent with the STEM images of V-doped 2H-TMDs[35–37]. As a result, the substitutional V atoms are at an average concentration of 4.0 at% (~2.0 wt%) in the 1T-WS₂ layer. To exclude the presence of V-based contaminations (e.g., V₂O₃, VO₂, and VS₂) in the transferred 1T sample, XPS spectra of 1T sample transferred on highly oriented pyrolytic graphite were performed (see details in Supplementary Figs. 10–12). The V signal in Supplementary Figs. 11c and 12c were ascribed to the V–S bond[38], which was consistent with the EELS result in Fig. 2e.

The cross-sectional view STEM image of V SACs@1T-WS₂ on the sapphire substrate was carried out to confirm the epitaxial relationship between V₂O₃ and the sapphire substrate. Figure 2g shows a low-resolution STEM image, indicating a ~20 nm thickness of V₂O₃ film. The inset high-resolution STEM image displays a sharp interface between V₂O₃ and sapphire substrate, revealing the epitaxial growth of V₂O₃ film on the sapphire substrate[39]. More importantly, we observe a single layer of WS₂

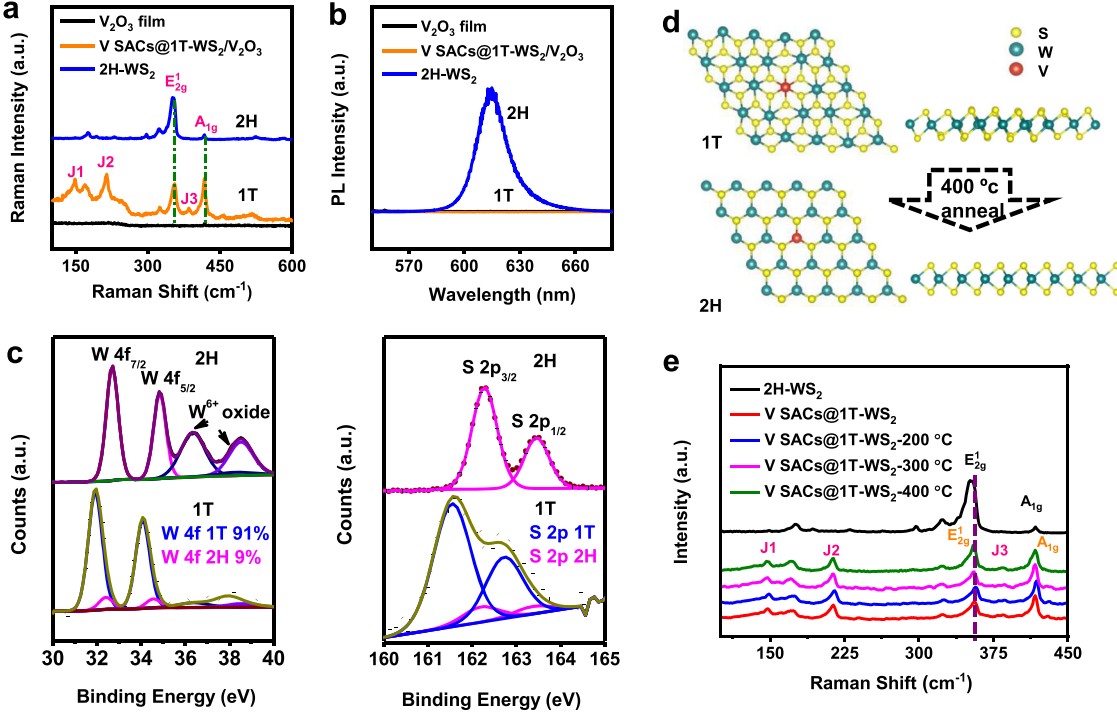

**Fig. 3 Phase transition from 1T to 2H. a** Raman spectra of $V_2O_3$ film (black), V SACs@1T-WS$_2$/V$_2$O$_3$ (orange), and 2H-WS$_2$ (blue). **b** PL spectra of $V_2O_3$ film (black), V SACs@1T-WS$_2$/V$_2$O$_3$ (orange), and 2H-WS$_2$ (blue). **c** High-resolution XPS spectra of W 4f (left) and S 2p (right) core level peak regions for 2H and 1T phase, respectively. The fitting blue and pink curves represent the contributions of 1T and 2H phases, respectively. **d** Schematic representation of the 1T to 2H structural phase transition occurring upon thermal annealing. **e** Raman spectra of 2H-WS$_2$, V SACs@1T-WS$_2$, and V SACs@1T-WS$_2$ with different annealing temperatures in H$_2$/Ar for 2 h.

from the sharp interface between V SACs@1T-WS$_2$ and V$_2$O$_3$ (Fig. 2h). The lattice spacing of V$_2$O$_3$ measured in the STEM image is ~0.23 nm, in correspondence with the (006) plane of V$_2$O$_3$ obtained from the XRD diffraction pattern in Supplementary Fig. 2[39]. In addition, the monolayer thickness of V SACs@1T-WS$_2$ (before and after transferring) is also verified by atomic force microscope (AFM), as shown in Supplementary Fig. 13. Second-harmonic generation and angle-resolved-polarization Raman spectra (ARPRS) are commonly used to probe the symmetry and isotropic/anisotropic lattice structure of TMDs[11,40,41]. Therefore, the high symmetry and isotropic nature of V SACs@1T-WS$_2$ have also been discussed in Supplementary Figs. 14 and 15.

**Controlled phase conversion**. The two different phases of 1T and 2H are easily highlighted by the confocal Raman spectroscopy, as provided in Fig. 3a, b. The Raman spectrum of V SACs@1T-WS$_2$ exhibits two prominent peaks at 418.4 cm$^{-1}$ and 354.6 cm$^{-1}$ (Fig. 3a, orange plot), corresponding to the A$_{1g}$ and E$^1_{2g}$ resonance modes, respectively. Moreover, three additional typical peaks are observed at 147.9 cm$^{-1}$ ($J_1$), 214.5 cm$^{-1}$ ($J_2$), and 385.3 cm$^{-1}$ ($J_3$), respectively, which only exist in the metallic phase but not in 2H-WS$_2$ monolayers (blue plot in Fig. 3a)[1,16,18,22,42–45]. Particularly, a prominently merged peak in the range of 150–250 cm$^{-1}$ can be ascribed to hexagonal V$_2$O$_3$, as demonstrated in Supplementary Fig. 16. As verified in Supplementary Fig. 17, V$_2$O$_3$ film starts to nucleate when the temperature is elevated at ~600 °C. Hence, the different nucleation temperatures of V$_2$O$_3$ and WS$_2$ give rise to the sequential growth of V$_2$O$_3$ and WS$_2$ on the sapphire substrate. A brief discussion of the possible formation mechanism of the V$_2$O$_3$ film on the sapphire substrate is provided in Supplementary Fig. 17. In addition, we can see that the strongest photoluminescence (PL) signal (Fig. 3b, blue plot) is associated with

monolayered 2H-WS$_2$[31], while the negligible PL signal (Fig. 3b and Supplementary Fig. 18, orange plot) from V@ SACs@1T-WS$_2$ is presumably due to the metallic nature[1,2].

To illustrate the high purity of the obtained 1T phase, XPS spectra were performed to quantify the 1T and 2H compositions according to the high sensitivity of the tungsten signal to its oxidation state and coordination geometry[1,46]. The V@ SACs@1T-WS$_2$ monolayers were transferred on the fresh sapphire substrate for the XPS measurements. The survey scan of V@ SACs@1T-WS$_2$ was shown in Supplementary Fig. 19a. The O 1s signal in Supplementary Fig. 19b was ascribed to sapphire substrate. As provided in Fig. 3c, the W 4f core level peak of 1T phase is shifted to lower binding energy of ~1.0 eV than that of 2H phase, assigning to a major amount as high as 91% of 1T phase. The decline in binding energy is presumably caused by the chemical reduction of W from +4 to the +3 oxidation state[47]. The existence of 2H phase in the 1T sample may arise from the phase transition caused by the X-ray illumination, as no 2H characters have been detected from Raman spectra and STEM images, implying that a very high phase purity of the as-grown 1T sample (at least 91%). Notably, signals at higher binding energies of 36.3 eV and 38.5 eV from the 2H-WS$_2$ stand for the peaks of W 5p$_{3/2}$ and W$^{6+}$ oxidation state species, respectively. Simultaneously, the high-resolution of core level S 2p peaks is coherently manifested lower binding energy in contrast to the S 2p peak from the 2H phase, consistent with the previous XPS studies of metallic WS$_2$[1,22].

Moreover, the V SACs@1T-WS$_2$ monolayers are extremely stable under ambient conditions even after one year, as shown in Supplementary Fig. 20. Importantly, a high 1T phase (~60%) is preserved in the sample from the XPS analysis in Supplementary Fig. 21b, and the decrease of 1T/2H ratio should be probably caused by the oxidation of V$_2$O$_3$ film on the surface, as

demonstrated in Supplementary Fig. 21d. In addition, upon annealing at different temperatures (200 °C, 300 °C, and 400 °C, respectively) in $H_2$/Ar condition, the 1T phase is partially transformed into a 2H phase. The schematic structure of phase transition from 1T to 2H is shown in Fig. 3d. From the XPS spectra recorded in Supplementary Fig. 22, despite gradual shrinkage of W 4f and S 2p peaks from 1T phase with the elevated temperature ≥300 °C, a very high amount of 1T phase is still preserved even after annealing at 400 °C (~49%). The different ratios of 1T/2H (W 4f) under different annealing temperatures were summarized in Supplementary Table 2. Raman spectra in Fig. 3e reveal that the enhanced intensities of $E_{2g}^1/A_{1g}$ are associated with the decreasing 1T phase. Remarkably, characteristic peaks assigned to the 1T phase are still observed after annealing at 400 °C. Interestingly, as the increasing temperature, the PL intensity in Supplementary Fig. 23 was enhanced and largely blue-shifted, which was ascribed to the variation of the band-structure caused by the pronounced ratio of 2H phase in the 1T sample, in consistent with the previously reported metallic TMDs[22].

**Growth mechanism of V SACs@1T-WS$_2$.** Our fabrication of V SACs@1T-WS$_2$ monolayers by the one-step growth exemplifies the advantage of the CVD strategy over the exfoliated method

and wet-chemical method, where the phase purity cannot be well controlled (Supplementary Table 1). The octahedral 1T phase has long been considered as an energetically unfavorable structure and tends to be transformed into a more stable 1T′ or 2H phase. Hence, understanding the growth mechanism of the present stable 1T structure is of importance for exploring more 1T-TMDs controllable growth. As demonstrated in Supplementary Fig. 17, the nucleation of $V_2O_3$ (at ~600 °C) precedes the epitaxial growth of WS$_2$ (~800 °C) during the whole growth. Stepwise-products experiments were designed to explore the origin of the V SACs@1T-WS$_2$ formation. The pristine one-step CVD growth was separated into two steps, that is initial $V_2O_3$ growth and subsequent V SACs@1T-WS$_2$ growth, as depicted in Fig. 4a.

In the first growth step, VCl$_3$ and sulfur were the co-precursors for the preparation of $V_2O_3$-nuclei film under the same growth condition with the pristine one-step growth. The presence of $V_2O_3$ was identified by the Raman spectrum in Supplementary Fig. 24a (blue plot) and XRD diffraction peaks in Supplementary Fig. 24b (blue plot). The oxygen element in the $V_2O_3$-nuclei film was presumably from the $O_2$ residue in the tube furnace, as $V_2O_3$-No nuclei film was also synthesized if using VCl$_3$ as the only precursor (Supplementary Fig. 24b, orange plot). Apart from the $V_2O_3$, trace amounts of VS$_2$ were also identified based on the XPS spectra analysis (Supplementary Fig. 24d, f). To further confirm the presence of VS$_2$, the $V_2O_3$-nuclei film was scraped

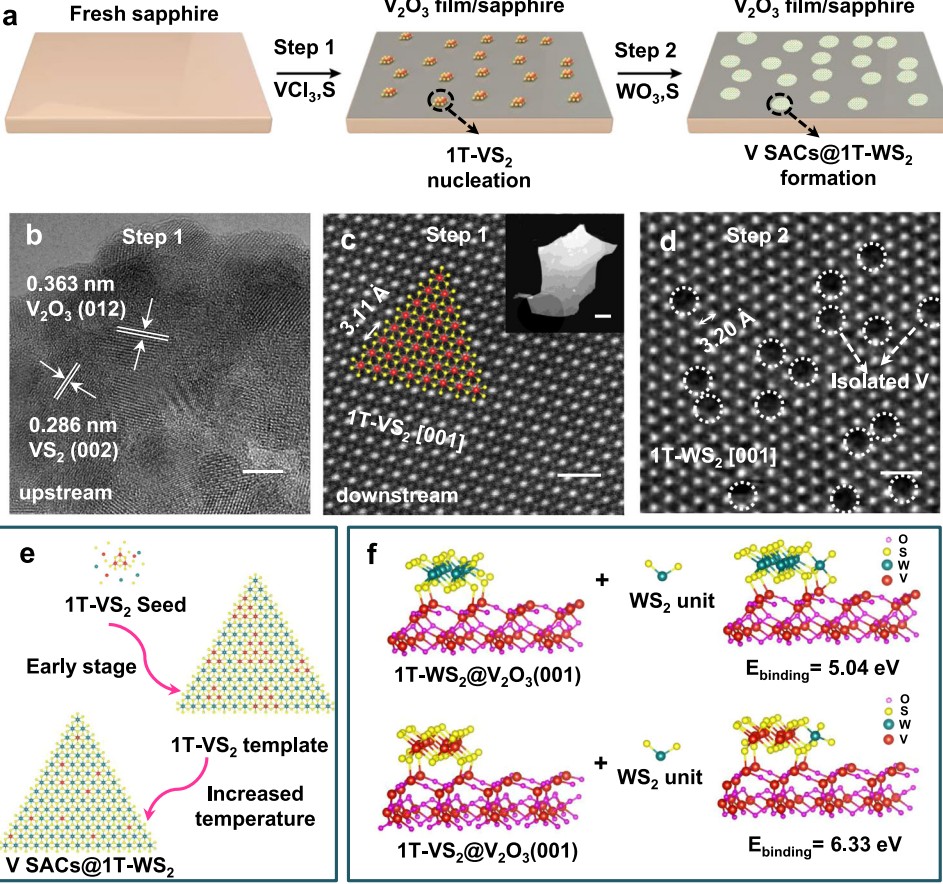

**Fig. 4 Growth mechanism of V SACs@1T-WS$_2$. a** Stepwise-products experiments for the V SACs@1T-WS$_2$ growth process. **b** HRTEM image of $V_2O_3$-nuclei film formed in the upstream of the sapphire substrate. **c** High-resolution HAADF-STEM image of 1T-VS$_2$ formed in the downstream of $V_2O_3$-nuclei film/sapphire substrate. The red spheres and yellow spheres represent the V atoms and S atoms, respectively, in the schematic model of 1T-VS$_2$ in the basal plane. Inset: low-resolution of HAADF-STEM image of VS$_2$ domain. **d** High-resolution HAADF-STEM image of V SACs@1T-WS$_2$ formed on the surface of $V_2O_3$-nuclei film. The V atoms are highlighted by the dashed white circles. **e** The proposed growth mechanism of V SACs@1T-WS$_2$ formed on the $V_2O_3$ film. **f** Computed binding energy of WS$_2$ unit on 1T-WS$_2$ vs 1T-VS$_2$ on the surface of $V_2O_3$ (001). Scale bars: (**b**) 5 nm; (**c**) 0.5 nm, inset 50 nm; (**d**) 0.5 nm.

from the sapphire substrate for the HRTEM measurement, as shown in Fig. 4b. In addition to the lattice spacing of 0.363 nm from $V_2O_3$, the other measured lattice spacing of ~0.286 nm can be well indexed with the (002) plane of 1T-$VS_2$. Elemental mapping from the same region showed the uniform distributions of V, O, and S elements (Supplementary Fig. 25) in the $V_2O_3$-nuclei film, which was coincident with the analysis of the XPS survey spectrum in Supplementary Fig. 24c. More interestingly, 1T-$VS_2$ nanocrystals were found on the surface of $V_2O_3$-nuclei film further downstream (Supplementary Fig. 26b, c, lower growth temperature zone, ~620 °C). In contrast to the trace amounts of $VS_2$ nuclei found in the upstream (the hot center growth zone, ~860 °C), the $VS_2$ nanocrystals downstream can be easily transferred from the $V_2O_3$-nuclei film using a standard transfer method for STEM imaging (Fig. 4c). The image showed a hexagonal atomic structure of 1T-$VS_2$ in the [001] basal plane[48], which was solid evidence that the $VS_2$ intermediates were nucleated on the surface of $V_2O_3$ during the growth.

In the second step, the as-produced $V_2O_3$-nuclei film/sapphire substrate was put back into a fresh CVD process to collect the final product using $WO_3$ and sulfur as the co-precursors. Intriguingly, it was found that smaller 1T domains of $WS_2$ were formed on the surface of the film (Fig. 4d, Supplementary Fig. 27a, e), whereas 2H-$WS_2$ domains were achieved if using $V_2O_3$-No nuclei film (Supplementary Fig. 27b, e, f) as the collecting substrate. Notably, nothing could be found if using commercial $V_2O_3$ (001) film (Supplementary Fig. 27c, d, and g), because that the rough surface and the low-quality of the commercial $V_2O_3$ were not in favor of the $VS_2$ and $WS_2$ nucleation on the surface. The 1T- or 2H-$WS_2$ domains were also confirmed by the Raman mapping in Supplementary Figs. 28 and 29. As a result, these experimental observations imply that both the $VS_2$ nucleation and the $V_2O_3$ film are very requisite for the V SACs@1T-$WS_2$ growth and $VS_2$ intermediates play the most important role in determining the 1T phase growth of $WS_2$. The proposed growth mechanism is shown in Fig. 4e, in which the

1T-$VS_2$ nuclei formed in the early growth stage serve as the 1T structure template and significantly strengthen the binding of $WS_2$ unit (Fig. 4f, from 5.04 eV to 6.33 eV) on the $V_2O_3$ film, thereby triggering the epitaxial growth of the 1T phase nucleus. Such an epitaxially grown $WS_2$ layer should be the 1T phase instead of the 2H phase due to the much higher binding energy of 1T-$VS_2$/1T-$WS_2$ (6.09 eV) than 1T-$VS_2$/2H-$WS_2$ (4.19 eV) (Supplementary Fig. 30). The computed lattice parameters of 2H-$WS_2$, V SACs@2H-$WS_2$, and V SACs@1T-$WS_2$ are shown in Supplementary Table 3. In addition, it was also demonstrated that the amount of $VCl_3$ could significantly affect the controllable phase growth of $WS_2$ (see details in Supplementary Figs. 31–34). The influence of heating temperature on the synthesis of 1T-$WS_2$ has also been investigated, as displayed in Supplementary Figs. 35–37. Moreover, vanadocene precursors were also investigated to enrich the growth method of 1T-$WS_2$ monolayers (Supplementary Fig. 38).

**HER activity of V SACs@1T-$WS_2$.** The as-produced V SACs@1T-$WS_2$ monolayers were transferred on the glass carbon (GC) electrode for the HER performance measurement using a three-electrode setup in 0.5 M $H_2SO_4$. In addition, HER performance of the other investigated electrocatalysts, i.e., bare GC, 2H-$WS_2$, 1T-400 (V SACs@1T-$WS_2$ annealed at 400 °C in $H_2$/Ar for 2 h), $2H_{1T}$ (transformed by V SACs@1T-$WS_2$ annealed at 200 °C in the air for 30 min) and commercial Pt/C-20 % were evaluated for comparison. As shown in linear sweep voltammetry (LSV) curves (Fig. 5a), V SACs@1T-$WS_2$ exhibited a low overpotential of 185 mV ($\eta_{10}$) at a current density of 10 mA/cm² with an ultralow mass loading of 1.8–6.5 μg/cm² (see details in Supplementary Figs. S39–S45 and Supplementary Table 4–6), outperforming the 1T-400 (blue plot, $\eta_{10}$ = 325 mV), $2H_{1T}$ (olive plot, $\eta_{10}$ = 515 mV) and 2H (red plot, $\eta_{10}$ = 684 mV) electrodes. The excellent activity of SACs@1T-$WS_2$ monolayers was further demonstrated by the comparisons of Tafel slopes for different catalysts, as shown in

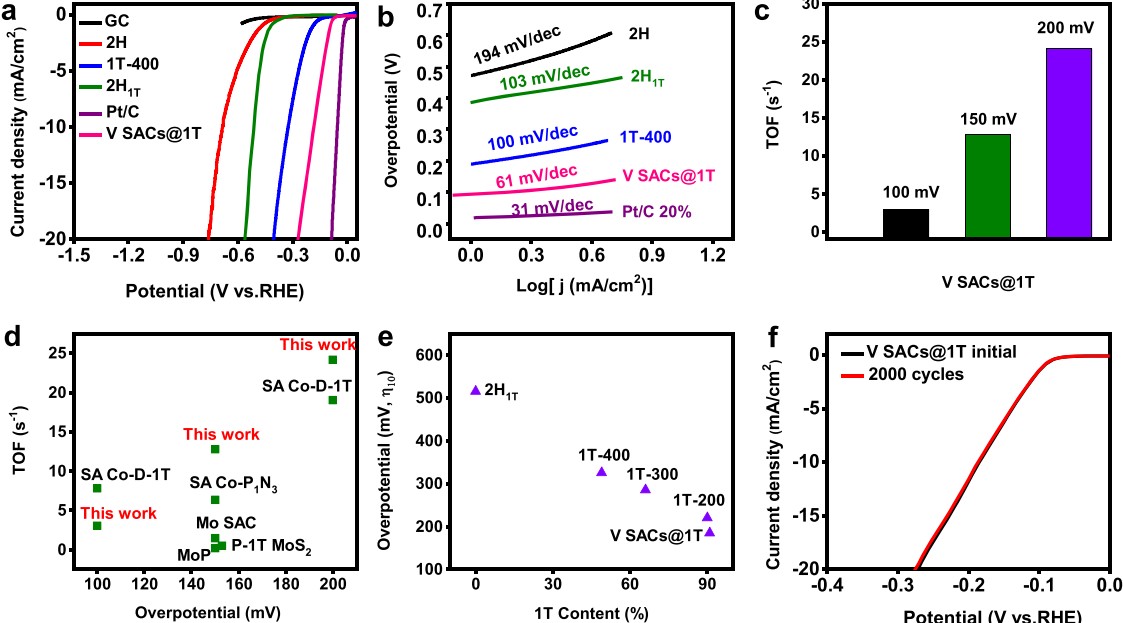

**Fig. 5 HER performance of V SACs@1T-$WS_2$. a** LSV curves of GC, 2H (2H-$WS_2$), 1T-400 (V SACs@1T-$WS_2$ annealed at 400 °C in $H_2$/Ar for 2 h), $2H_{1T}$ (V SACs@2H-$WS_2$, transformed by V SACs@1T-$WS_2$), Pt/C (Pt/C 20%) and V SACs@1T-$WS_2$ electrodes in 0.5 M $H_2SO_4$ with a scan rate of 5 mV/s. **b** Tafel plots of 2H, 1T-400, $2H_{1T}$, Pt/C 20% and V SACs@1T electrodes. **c** TOF values of V SACs@1T-$WS_2$ measured at different overpotentials. **d** TOF comparison with other representative electrocatalysts. **e** HER activity as a function of the 1T phase ratios obtained by annealing V SACs@1T-$WS_2$ in the $H_2$/Ar for 2 h. **f** Electrochemical stability of V SACs@1T-$WS_2$ after 2000 cycles of CV tests.

Fig. 5b. The Tafel slope for V SACs@1T-WS₂ was 61 mV/dec (Fig. 5c), which is much lower than that of 1T-400 (100 mV/dec), $2H_{1T}$ (103 mV/dec), and 2H-WS₂ (194 mV/dec). Please note that mass loading of electrocatalysts has a great impact on the measured activities; hence, it has generally used as catalyst performance metrics[7]. Although the mass loading of SACs@1T-WS₂ was much lower than most of the non-precious electrocatalysts (Supplementary Fig. 46), the HER performance was still comparable to that of strained 1T′-WS₂ nanosheets (NSs)[1], 1T′-MoS₂ NSs[2], 1T′-MoS₂ monolayers[11], and better than that of 2H-1T boundaries MoS₂ monolayers[49], 1T′ WSe₂ NSs[7] and PE-CVD 1T-WS₂ film[15]. The electrochemical surface area (ECSA) was calculated as an important factor to affect the catalytic activity of electrocatalysts[50–52]. The calculated details were shown in the experimental section and Supplementary Fig. 47. Double-layer capacitance ($C_{dl}$) and ECSA values were 139.5 μF/cm² and 3.49 cm² for V SACs@1T-WS₂ and 61.7 μF/cm² and 1.54 cm² for 2H-WS₂, suggesting the critical contributions of V atom sites and high purity of 1T-WS₂. The TOF values of the SACs@1T-WS₂ (Fig. 5c) were obtained according to the precious reports[23,53] and calculated to be 3.01 s⁻¹, 12.78 s⁻¹, and 24.15 s⁻¹ at overpotentials of 100 mV, 150 mV, and 200 mV, respectively, which were much higher than those recently representative electrocatalysts listed in Fig. 5d and Supplementary Table 7.

As the V SACs@1T-WS₂ monolayers contain a high concentration of 1T phase, to investigate the influence of 1T content on the catalytic performance, the V SACs@1T-WS₂ electrode was gradually annealed at different temperatures and the HER activity after each annealing treatment was performed, ensuring that the identical dimensions and geometrical areas to evaluate the catalytic properties. Remarkably, the HER activities were strongly related to the 1T phase content as demonstrated by a gradual decrease in the $\eta_{10}$ with decreasing 1T ratios (Fig. 5e). Remarkably, negligible activity degradation can be observed in the reproducible polarization curve of V SACs@1T-WS₂ in Fig. 5f even after 2000 continuous cycling, indicative of a good HER performance stability. The stability test was conducted at current densities higher than 10 mA/cm² in 0.5 M H₂SO₄ electrolyte for 100 h. As revealed by the chronoamperometric curve of V SACs 1T-WS₂ electrocatalyst in Supplementary Fig. 48, the current density for the V SACs 1T-WS₂ electrocatalyst displayed a slight current decay of 1.0 mA cm⁻² after 24 h and 3.4 mA cm⁻² after 100 h, indicating high stability of V SACs 1T-WS₂ catalyst. The metallic properties of V SACs 1T-WS₂ catalyst after stability test were also investigated by Raman spectroscopy (Supplementary

Fig. 49a), which showed obvious metallic peaks ($J_1$, $J_2$, $J_3$) in the Raman spectrum (red plot). Moreover, the STEM image showed in Supplementary Fig. 49b confirmed the V SACs 1T-WS₂ structure after the stability test. Both the Raman spectrum and STEM image indicated the robust 1T structure of V SACs 1T-WS₂ catalyst after HER test.

**V SACs enhancing the HER activity of 1T-WS₂.** Especially, the HER performance of 2H-WS₂, V SACs@2H-WS₂, and V SACs@1T-WS₂ was further studied using DFT considering both basal plane (Supplementary Fig. 50) and edge sites (Supplementary Figs. 51–53) as the active sites. Our calculations show that the V SACs could significantly influence the free energy of H adsorption ($\Delta G_H$) on the edge sites of 1T-WS₂, as summarized in Supplementary Table 8. Please note that the $|\Delta G_H|$ of V SACs@1T-WS₂ at the basal plane is 0.4 eV, which is approximate to the $|\Delta G_H|$ of intrinsic 1T-WS₂ at 0.28 eV[1], implying that the single-atom V sites have a negligible influence on the HER performance of 1T-WS₂ in the basal plane. Compared to the 2H-WS₂ and V SACs@2H-WS₂, V SACs@1T-WS₂ exhibits the lowest $|\Delta G_H|$ (0.05 eV) at V-atom sites (Fig. 6a), indicating that the isolated V atoms are catalytically active in the layer of 1T-WS₂. The charge depletion at the active sites has been proven to play an essential role in improving the electrochemical activity of the catalysts[13,54]. To acquire a deeper understanding of how single V atoms enhanced the activity of 1T-WS₂, the charge redistribution of V SACs@1T-WS₂ was studied. As shown in the inset image in Fig. 6b, we can clearly see that when one W atom was substituted by a V atom, there was more charge depletion generated at the V-atom site. Such variations in the local electronic structure can also be well described using the d-band theory[55]. A linear inverse correlation between $|\Delta G_H|$ and the d-band center at the most active edge sites were revealed, indicating that a more negative d-band center corresponds to more occupation of the antibonding states, resulting in weaker H adsorption of V SACs@1T-WS₂ comparable to their 2H counterparts.

**Discussion**

In summary, we have demonstrated a direct synthesis of single-atom V sites on the high purity 1T-WS₂ monolayers via a one-step CVD strategy through introducing VCl₃ as a co-precursor, with much better HER performance than the 2H counterparts. The step-wise experimental findings together with DFT results shed light on the understanding of the CVD-grown V SACs@1T-WS₂ monolayers, that is the 1T-VS₂ nuclei initially act as the 1T template structures for

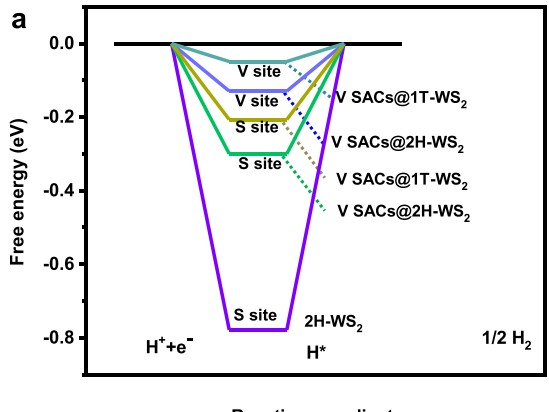
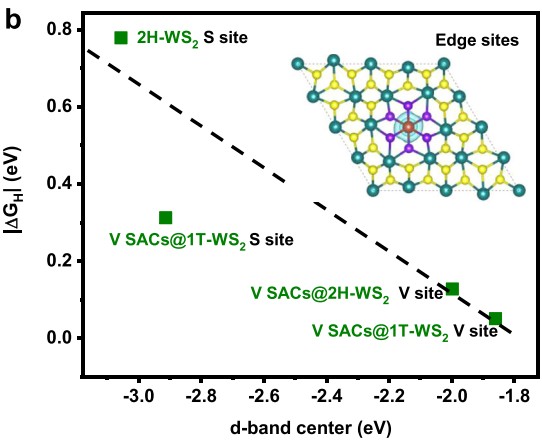

**Fig. 6 HER activity mechanism of V SACs@1T-WS₂. a** The Gibbs free energy of H adsorption ($\Delta G_{H^*}$) of 2H-WS₂, V SACs@2H-WS₂, and V SACs@1T-WS₂ on the V and S edge sites. **b** Relationship between $|\Delta G_{H^*}|$ and the d-band center of adjacent W/V ions at the most active edge sites of 2H-WS₂, V SACs@2H-WS₂, and V SACs@1T-WS₂.

the follow-up V SACs@1T-WS$_2$ growth on the V$_2$O$_3$ film. The HER performance of intrinsic 1T-WS$_2$ was remarkably boosted after the modification of isolated V atoms. DFT calculations highlight that the highly activated V-atom sites primarily account for the excellent HER activity of V SACs@1T-WS$_2$ monolayers. Our findings could fill the gap of SACs grown via a one-step CVD strategy and pave the way to design more efficient and robust electrocatalysts for applications in water splitting.

## Methods

**The growth of V SACs@1T-WS$_2$ monolayers, 2H-WS$_2$ monolayers, V$_2$O$_3$ film.** In the growth of monolayer V SACs@1T-WS$_2$, sulfur (Sigma-Aldrich, 99%), tungsten trioxide (WO$_3$, Sigma-Aldrich, 99.99%), and vanadium (III) chloride (VCl$_3$, Sigma-Aldrich, 99%,) were used as the growth precursors. Two quartz boats with 0.03 g of VCl$_3$ powder (in the hot center) and 0.30 g WO$_3$ powder (the upstream of VCl$_3$) were placed sequentially in the hot center of the furnace. And a fresh sapphire (c-plane) substrate (1 cm × 2 cm) was placed face-down of the quartz boat to collect the final product. The heating temperature was kept at 860 °C. The S powder was placed in a separate quartz boat at the upper stream side of the cold zone and the temperature was maintained at 160 °C during the reaction. The gas flow has consisted of an ultrapure mixed Ar/H$_2$ (Ar = 80 sccm, H$_2$ = 20 sccm), and the chamber pressure was controlled at 10 Torr. After reaching the desired growth temperature of 860 °C, the heating zone was kept for 15 min and the furnace was then naturally cooled down to room temperature. The V$_2$O$_3$ film was simultaneously formed during the V SACs@1T-WS$_2$ growth. The commercial V$_2$O$_3$ (001) film on the c-plane sapphire was purchased from Xi'an Qiyue Biotechnology Co., Ltd. To obtained 2H-WS$_2$ monolayers, we used WO$_3$ and suffer as the precursors with the same growth condition with V SACs@1T-WS$_2$. To obtain V$_2$O$_3$-No nuclei film, we used the only VCl$_3$ as the precursors with the same growth condition. To obtain V$_2$O$_3$-nuclei film, we used only VCl$_3$ and sulfur as the precursors with the same growth condition.

**V$_2$O$_3$ film growth mechanism and characterization.** According to the lattice parameters of V$_2$O$_3$ and sapphire (V$_2$O$_3$: hexagonal, $a$ = 0.492 nm, $c$ = 1.397 nm; α-Al$_2$O$_3$: hexagonal, $a$ = 0.475 nm, $c$ = 1.297 nm), hexagonal V$_2$O$_3$ shares the same lattice-type and similar lattice parameters with α-Al$_2$O$_3$; thus, it is expected to epitaxially grow on α-Al$_2$O$_3$ with the relationship V$_2$O$_3$ (001)[100]//α-Al$_2$O$_3$ (001)[100][39,56–58].

To investigate the growth mechanism of V$_2$O$_3$ film, we prepared V$_2$O$_3$-No nuclei film under different temperatures, as exhibited in Supplementary Fig. 17a. It can be observed that V$_2$O$_3$ film could be formed at ≥600 °C during the temperature-dependent controllable experiments. Particularly, a merged peak in the range of 150–250 cm$^{-1}$ was observed in the Raman spectrum and the peak was correlated to both monoclinic A$_{1g}$ and hexagonal V$_2$O$_3$ A$_{1g}$ symmetry, indicative of a mixed phase. However, the additional peaks of low intensity at 300 cm$^{-1}$ and 500 cm$^{-1}$ were solely attributable to hexagonal V$_2$O$_3$[59,60], indicating the hexagonal structure of the as-grown V$_2$O$_3$ film. Under the temperature of 600 °C, low-quality film or nothing will be formed on the sapphire substrate. Simultaneously, we measured the Raman spectra of the powders from the VCl$_3$ quarts boat after each growth reaction with different growth temperatures, as shown in Supplementary Fig. 17b. VO$_2$ (B) will be evidently formed after the decomposition of VCl$_3$[61] (see Eqs. (1) and (2) in Supplementary Fig. 17).

**Transfer process of V SACs@1T-WS$_2$, 2H-WS$_2$, V$_2$O$_3$-nuclei film.** The as-grown samples were transferred onto arbitrary substrates, such as fresh sapphire, SiO$_2$/Si, GC, and holy-carbon nickel TEM Grid using a modified method in our lab. First, poly (methyl methacrylate) (PMMA) was spin-coated onto the samples followed by 120 °C baking for 10 min, and then etched by 5% HF solution for 2–3 min. The PMMA/ sample was gently peeled off by the tweezers and deposited onto the targeted substrates. PMMA/sample/targeted substrate was baked for 1 h at 120 °C in air. The PMMA was removed by acetone and cleaned with isopropyl alcohol. The V$_2$O$_3$-nuclei film was scratched by tweezers and ultrasonic the V$_2$O$_3$/sapphire sample in the ethanol solution for 15 min for the HRTEM measurement. The sample was prepared by dropping the solution onto the holy-carbon nickel TEM grid.

For the STEM measurements samples, the extra annealing process was necessary to remove the residual PMMA. For the 2H-WS$_2$ sample annealing, the sample was located in a vacuum container (less than 10$^{-6}$ Torr) at 350 °C overnight. For the V SACs@1T-WS$_2$ sample annealing, the sample was located in a vacuum container (less than 10$^{-6}$ Torr) at 200 °C for 2 h. For the XPS and Raman spectra of V SACs@1T-WS$_2$ annealed at different temperatures, the V SACs@1T-WS$_2$ monolayers were transferred on the fresh sapphire substrates and annealed in H$_2$/Ar condition with different temperatures.

**Characterization.** Optical spectroscopy is collected under a Witec alpha 300 R confocal Raman microscopic system. Gratings of 1800 lg/mm and 300 lg/mm are selected for the high-resolution Raman spectrum and wide range PL spectrum, respectively. The TMDs are excited by 532 nm laser with a power of 1 mW and

spot size of 0.5 μm and emitted Raman signal is collected by 100× objective (N.A = 0.9) from a Carl Zeiss Microscopy. ARPRS are conducted on a triple-grating micro-Raman spectrometer (Horiba-JY T64000) with a 532 nm laser under a backscattering configuration. The polarization is resolved by rotating sample orientation on the normal axis of the basal plane. The emitted Raman signal is collected through a 100× objective, dispersed with 1800 lg/mm grating, and detected by a charge-coupled device.

An AFM (Cypher ES environmental AFM) was utilized to obtain the morphology images and the height profile of V SACs@WS$_2$ monolayers. The crystalline of the different samples was probed using a Bruker D8 advance powder XRD with Cu Ka radiation. X-ray photoelectron spectroscopy (XPS) studies were carried out in a Kratos Axis Ultra DLD spectrometer equipped with a monochromatic Al Kα X-ray source ($hv$ = 1486.6 eV) under a vacuum of 1 × 10$^{-9}$ mbar. The spectra were collected at fixed analyzer pass energies of 160 eV and 20 eV. The binding energies in XPS analysis were corrected by referencing C 1s line at 284.8 eV. STEM and TEM images were performed by Titan 40-300 Themis Z TEM from Thermo Fisher, USA (former FEI Co) equipped with a double Cs corrector, an electron monochromator, and a Gatan imaging filter quantum 966. The microscope was operated at 80 kV to minimize electron beam induced. Probe semi convergence angle was tuned to 30 mrad and probe current to 50 pA. For high-angle annular dark-field (HAADF) STEM images the inner collection angle was about 80 mrad. Radial wiener filter was carried out to enhance the visibility of atoms

**Computational details.** First-principles simulations were performed at the DFT level implemented by the Vienna ab initio simulation package[62–65] with the all-electron projector augmented wave model[66] and Perdew–Burke–Ernzerhof functions[67]. The DFT-D$_3$ method was applied to include vdW interaction corrections[68]. A kinetic energy cutoff of 400 eV was used for the plane-wave expansion of the electronic wave function. The convergence criteria of force and energy were set as 0.01 eVÅ$^{-1}$ and 10$^{-5}$ eV, respectively. Gaussian smearing of 0.1 eV was applied for optimization. A k-point grid with a 4 × 4 × 1 gamma-centered mesh was used for the WS$_2$ unit cell. For supercells that contain a larger number of vanadium and oxygen atoms, a corresponding number of k-points were used to keep the k-mesh spacing constant across different structures. The climbing image nudged elastic band was applied for transition state optimization[69].

The free energy of the adsorption atomic hydrogen ($\Delta G_H$) is obtained by $\Delta G_H = \Delta E_H + \Delta E_{ZPE} - T\Delta S_H$. $\Delta E_H$ is the adsorption energy defined by $\Delta E_H = E_{sur} - H - E_{sur} + \frac{1}{2} E_{H2}$. According to previous reports, $\Delta G_H$ can be written as $\Delta G_H = \Delta E_H + 0.25$, where 0.25 eV is the contribution from ZPE and entropy at 298 K[70].

**Electrochemical measurements.** Electrochemical measurements were carried in a PGSTAT 302N Autolab Potentiostat/Galvanostat (Metrohm) at room temperature. Graphite rod and Ag/AgCl (in a saturated KCl solution) electrodes were employed as the counter and reference electrodes, respectively. The V SACs@1T-WS$_2$ and 2H-WS$_2$ were transferred onto the GC electrodes as the working electrodes and dried. Nafion solution (0.5%) was drop-cast to protect WS$_2$ film. The V SACs@1T-400 electrode was prepared by annealing the V SACs@1T-WS$_2$ electrode at 400 °C in H$_2$/Ar for 2 h. The 2H$_{1T}$ electrode was prepared by annealing the V SACs@1T-WS$_2$ electrode at 200 °C in the air for 30 min. The HER activities of different samples were evaluated by measuring polarization curves with LSV at a scan rate of 0.5 mV/s in 0.5 M H$_2$SO$_4$ solutions. Potentials were referenced to a reversible hydrogen electrode (RHE). The commercial Pt/C (20 wt% Pt on Vulcan carbon black) supported by GC was prepared by mixing the Pt/C, nafion, and isopropanol, sonicating for 30 min, and drop-casting on the GC. The mass loading of Pt/C was 500 μg/cm$^2$. The potential cycling was performed between 0.197 and −0.6 V vs RHE at 5 mV s$^{-1}$. All data have been corrected for a small ohmic drop based on impedance spectroscopy. ECSA and C$_{dl}$ are determined by cyclic voltammograms at various scan rates (10, 30, 50, 70, 90, 110. 130, 150, 170, 190, and 210 mV/s) in the potential range (0.15–0.35 V vs. RHE). The capacitive currents ($\Delta J$) are plotted as a function of scan rate and C$_{dl}$ is equal to half of the slope. The reference specific capacitance ($C_s$) of 40 μF/cm$^2$ is used in this work. The ECSA for the different catalysts are achieved based on the following equation

$$\text{ECSA} = \frac{C_{dl}}{C_s}$$

**Calculation of turnover frequency.** The TOF calculation details were specified as below, which was reported elsewhere[23,53].

$$\text{TOF} = \frac{\text{Total hygrogen turnovers per geometric area}}{\text{active sites per geometric area}}$$

The total hydrogen turnovers were calculated from the current density in the LSV polarization curve according to the equation as below:

$$\text{Total hydrogen turnovers} = \left(|j| \frac{mA}{cm^2}\right)\left(\frac{1C/s}{1000\ mA}\right)\left(\frac{1\ mol\ e^-}{96485\ C}\right)\left(\frac{1\ mol}{2\ mol\ e^-}\right)\frac{(6.022 \times 10^{23} \text{moleculars H}_2)}{1\ mol\ H_2}$$

The number of active sites in the V SACs@1T-WS$_2$ catalyst was obtained from the mass loading on the glass carbon electrode.

$$\text{Active sites} = \left( \frac{\text{electrocatalyst loading per geometric area}\left(\frac{g}{cm^2}\right) \times V\text{wt\%}}{VM_W\left(\frac{g}{mol}\right)} \right) \left( \frac{6.022 \times 10^{23} \text{V atoms}}{1 \text{ mol V}} \right)$$

## Data availability

The data that support the findings of this study are available from the corresponding author upon request.

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

## Acknowledgements

L.L. acknowledges support from King Abdullah University of Science and Technology (Saudi Arabia). Y.L. and D.W. acknowledge support from the National Key R&D Program of China (2018YFA0702003) and the National Natural Science Foundation of China (21890383, 21671117, 21871159). W.Z. acknowledges support from the Educational Commission of Guangdong Province project (No.2015KGJHZ006).

## Author contributions

Y.L., L.L., and D.W. conceived the idea and designed the research project. A.H. designed the synthesis and performance experiments, collected and analyzed the data, and wrote the manuscript. X.Z. contributed to the characterizations of samples and wrote the manuscript. X.W. contributed to the computational results and wrote the manuscript. S.L. designed the optical characterizations experiments, analyzed the data, and wrote the manuscript. L.G. and Q.Z. designed the simulation STEM analyzes. Q.X., Z.Z., F.L., W.Z. contributed to revising the manuscript. All the authors commented on the manuscript and have given approval to the final version of the manuscript.

## Competing interests

The authors declare no competing interests.
