## [Peer Review File · Nature Communications]

REVIEWER COMMENTS

Reviewer #1 (Remarks to the Author):

Authors have developed a novel HER electrocatalyst via a one-step chemical vapor deposition, i.e., single-atom-V substitution in 1T-WS₂ monolayers. It was observed that the 1T-VS₂ intermediates dominated the growth of the metallic WS₂ 1T phase. Due to the high loading of single-atom V substitutions the as-engineered electrocatalyst has demonstrated significantly enhanced HER activity, originated mainly from the activated V-atom sites. Generally, the manuscript was well organized and presented, and the conclusions are basically supported by the experimental data. Furthermore, as the authors stated, this study may open new avenue for direct synthesis of atomically dispersed single-metal catalysts on metastable materials as efficient and robust electrocatalysts for the HER and/or other electrocatalytic reactions. Therefore, I think the manuscript merit the publication after addressing the following issues.

- 1) Abstract: Two different "1T" materials (i.e., 1T-WS₂ monolayers and 1T-VS₂ intermediates) were mentioned in the sentence starting with "the growth mechanism", the reviewer is suggesting that "metallic WS₂" should be added immediately before the term "1T phase growth"
- 2) Abstract: No need to define "Density functional theory" as DFT because after the definition, it has not been used even once.
- 3) Introduction: Hydrogen generation from water splitting is an intriguing research topic, and various strategies have been explored, including photocatalytic HER, photoelectrochemically catalytic HER, and electrocatalytic HER. Therefore, a brief survey of recent literature closely related to these strategies is highly desired to enrich the research background, for example, g-C₃N₄-based nanocomposites (Energy Environ. Sci. 2019, 12, 2080-2147) for photocatalytic HER; Ag-Based nanocomposites (Nanoscale, 2019, 11, 7062-7096) for electrocatalytic and photocatalytic HER, etc.
- 4) Figure 3b, title for X axis: Please change "wavelenth" to "wavelength".
- 5) Supplementary Figure 1, caption: please delete the description "The blue and yellow balls represent ..." because these balls and their meanings have been illustrated in the schematic models.
- 6) Supplementary Fig. 10, caption: a) please change "different grown temperature" to "different grown temperatures", b) please change "was probably reaction" to "will probably react". Besides, please also change "x" to italic in the equation 1 to avoid any confusion.
- 7) Supplementary Fig. 12: Please change "fresh prepared" to "freshly prepared"
- 8) Supplementary Fig. 13, Caption: The authors described "Please note that the 1T sample was annealed at different temperatures in H₂/Ar for 2h". Does it mean that the same 1T sample was subjected to a series of heat treatments at different temperatures or three 1T samples were subjected to heat treatments at different temperatures? In the later case, please change "1T sample was" to "1T samples were".
- 9) Supplementary Figure 14b: Please use the same color for the same sample shown in Supplementary Figure 14a to avoid any confusion. In addition, in the caption: please change "partially 1T phase transformation" to "partial 1T phase transformation"
- 10) Supplementary Figure 15c-f: The values in X axis in Figure c show a decrease style from 1200 to 0, while the values in X axis in Figures e-f start from a lower one. Please make changes to make sure the values in X axis in these figures show a similar trend (decreasing or increasing).
- 11) Supplementary Figure 16a-d: The images for the EDX mapping are little blurry. The authors are suggested to provide images with a higher resolution. In addition, scale bars are also suggested to provide for these images.
- 12) Supplementary Figure 17a-b: scale bears were provided but without any description(s).
- 13) Supplementary Figure 18a-d: scale bears were provided but without any description(s).
- 14) Supplementary Figure 19b, caption: It would be better to change "Binding energy of 1T-VS₂/2H-WS₂ and 1T-VS₂/1T-WS₂" to Binding energy of 1T-VS₂/1T-WS₂ and 1T-VS₂/2H-WS₂"
- 15) Supplementary Figure 20, caption: It would better to change "HER performance comparison with different mass loadings of catalysts" to "HER performance comparison for the various catalysts with different mass loadings (the catalysts include: ..."
- 16) Supplementary Figure 20, Y axis: please change "slop" to "slope".

- 17) Supplementary Figure 21: what do they stand for by the different balls shown in the model structures?
- 18) Table S1: please change "Synthesized strategy" to "synthesis strategies", and "Ref." to "References"
- 19) Table S4: please change "Ref." to "Reference"

Reviewer #2 (Remarks to the Author):

Currently, it remains a huge challenge to directly synthesize 1T-WS₂, since it has the highest formation energy when compared with other transition metal dichalcogenides (TMDs) from Group VI elements with 1T phase. In this work, the authors have implemented a facile VCl₃-assist chemical vapor deposition (CVD) strategy to introduce atomically dispersed V-atoms onto 1T-WS₂ monolayers, which is a very original and heuristic work. Impressively, the authors have performed the detailed experimental and theoretical investigations on the growth mechanism and HER catalytic activity of V SACs on 1T-WS₂ monolayers. I highly recommend this work to be published on the Nature Communications. However, the following questions should be addressed before its publication:

- (1) In this work, the authors have prepared the 1T-WS₂ monolayers with the help of VCl₃, as 2H-WS₂ monolayers were obtained without VCl₃. This point should be quite interesting. However, I am wondering if the amount of VCl₃ has a great impact on the purity of 1T-WS₂.
- (2) Is the substrate of c-plane sapphire play an import role in the controllable phase growth? It is well-known that 2H-WS₂ monolayers are also able to epitaxially grow on the other substrate, such as SiO₂/Si wafer. Hence, if the authors use the same growth conditions, is it possible that the 1T phase samples are formed on the SiO₂/Si wafer or other kinds of substrates? The authors need to provide more information about the choice of substrate.
- (3) According to the author's statement, the V SACs on the 1T-WS₂ monolayers showed stable HER performance, however, there was no convincing evidence to exhibit the stability of the structure after HER test. Here, it is suggested that the authors add some tests, such as Raman, STEM, to show the structure after catalysis.
- (4) A technical question: in the AFM image of Supplementary Fig. 7a for demonstrating the thickness of WS₂ monolayer, why the topographic image was so blurry? Hence, it was very hard to clearly identify the morphology of WS₂.
- (5) Although the author has cited the paper for the calculations of TOFs, please specify the details for the TOFs in this work.
- (6) The catalytic tests to demonstrate the stability of the catalysts are incomplete. Evidently, the authors have shown the stability of catalysts after continuous CV cycles. However, this is not enough. Please carry out the stability test of long-term electrolysis (at least 10 h).

Reviewer #3 (Remarks to the Author):

In this work, the authors demonstrated a one-step CVD growth of V-doped 1T-WS₂. The authors also proposed a 1T-VS₂ nucleation and templated growth model, and discussed the importance of V₂O₃ substrate for 1T-WS₂ growth. Furthermore, HER catalysis properties of the V SACs@1T-WS₂ was tested, exhibiting enhanced HER properties compared to undoped 1T-WS₂. This is an interesting piece of work containing both experimental and theoretical efforts. However, my main concern is the effects of V₂O₃. It seems the formation of V₂O₃ underneath the material of interest is unavoidable, and it can have influence on different characterization techniques (e.g. EELS, XPS, HER properties measurement). It is also unclear to me whether the V₂O₃ is removed during transfer. In addition, considering the fact that V compounds can be easily oxidized when exposing to air, and the authors did not mention that all the characterizations were carried out without exposing them to air, thus the

characterizations and proposed mechanisms seem not reliable. The manuscript is also hard to read in some sections. Thus, I would recommend a rejection of the current manuscript, but if the authors address all the points, the paper could be eventually published.

Details are described below:

1. In Fig. 2f and Line 116-118 of the main manuscript, EELS was performed to identify V dopants. However, since there is a V₂O₃ film underneath the V SACs@1T-WS₂, signals of V can also from the V₂O₃. Can the authors comment on how to rule out this possibility?
2. Was any image filtering process used for enhancing the visibility of atoms in Fig. 2c and d? This should be included in experimental details.
3. In Line 155-157 of the manuscript, please check the resolution of the spectrometer to make sure all digits are significant. For example, "418.44" might be changed to "418.4".
4. For the phase purity analysis (Fig. 3c), if I understand correctly, the XPS comparison was done between pure 2H-WS₂ and V SACs@1T-WS₂/V₂O₃. I think the XPS peak shift can also originate from the interaction with V₂O₃, and also the effect of V doping, which can not be ruled out in this case, and makes the phase purity analysis not very convincing.
5. It seems the HER measurement was done on the V SACs@1T-WS₂/V₂O₃. Thus, similar to the previous question, could V₂O₃ contribute and enhance the measured catalytical performance?
6. Figure resolution need to be improved, and the label font size need to be the same.
7. Could the authors provide the V 2p XPS? Did the authors consider the V-S bond when fitting the V 2p peak?
8. In Figure S18, the 1T or 2H WS₂ domains are not clear. Could the authors label them? Raman mapping is recommended, so that the WS₂ region can be clearly separated out. In figure s18c, we could also see some domain with different contrast, so it would be better to use Raman mapping to identify the phase and the phase distribution.
9. The sample labeling may be a bit confusing. For example, from line 351 to 352:
"To obtain pure V₂O₃ film, we used only VCl₃ as the precursors with the same growth condition. To obtain V₂O₃ film (no W), we used only VCl₃ and sulfur as the precursors with the same growth condition."
It is a bit difficult to understand the difference between "pure V₂O₃ film" and "V₂O₃ film (no W)" without referring to this sentence, and it may cause misunderstanding when reading the article.
10. The authors tried to propose a growth mechanism in which the formation of V₂O₃ film and VS₂ domains are important for the 1T WS₂ growth. However, it is difficult to tell if the V₂O₃ is formed during the synthesis or during the characterization, as the V compounds (VS₂, VCl₃) are extremely air sensitive. If the characterization was done in air, oxidation can not be avoided, thus it may lead to misunderstanding in the mechanism study.
11. The authors claim that it is a controllable approach to synthesize 1T WS₂. Regarding the controllability, could the author control the 2H/1T ratio through the synthesis, not post annealing?
12. In addition, this approach relies on the introduction of V and the formation of 1T-VS₂ and V₂O₃ based on the proposed mechanism, could the other V precursors, such as vanadocene, give the same effect? What will happen if you change the VCl₃ amount? Do you have the tunability towards the V concentration without changing the 1T character of WS₂?

13. For the HER measurements, since the VWS2 is not a continuous film, how did the authors calculate the mass loading and the area? It seems that the author assume the film is a continuous film, which abbeys the characterization shown in Figure 2.

Responses to the Reviewers' Comments

We thank the reviewers very much for their valuable comments to our manuscript. We have carefully considered the reviewers' comments and revised the manuscript accordingly. Our responses and corresponding revisions are as follows:

Response to Reviewer #1

Comment: *Authors have developed a novel HER electrocatalyst via a one-step chemical vapor deposition, i.e., single-atom-V substitution in 1T-WS₂ monolayers. It was observed that the 1T-VS₂ intermediates dominated the growth of the metallic WS₂ 1T phase. Due to the high loading of single-atom V substitutions the as-engineered electrocatalyst has demonstrated significantly enhanced HER activity, originated mainly from the activated V-atom sites. Generally, the manuscript was well organized and presented, and the conclusions are basically supported by the experimental data. Furthermore, as the authors stated, this study may open new avenue for direct synthesis of atomically dispersed single-metal catalysts on metastable materials as efficient and robust electrocatalysts for the HER and/or other electrocatalytic reactions. Therefore, I think the manuscript merit the publication after addressing the following issues.*

Response: We appreciate the reviewer very much for the positive comment. His/Her concerns are addressed as follows.

Comment 1. Abstract: *Two different "1T" materials (i.e., 1T-WS₂ monolayers and 1T-VS₂ intermediates) were mentioned in the sentence starting with "the growth mechanism", the reviewer is suggesting that "metallic WS₂" should be added immediately before the term "1T phase growth"*

Response 1. We appreciate the reviewer very much for the helpful suggestion. According to your suggestion, we have added "metallic WS₂" before the term "1T phase growth" in the abstract section, which has been highlighted on Page 2 in the revised manuscript.

Comment 2. Abstract: No need to define “Density functional theory” as DFT because after the definition, it has not been used even once.

Response 2. We appreciate the reviewer very much for the suggestion. We agree with the reviewer that it is not necessary to define “Density functional theory” if it was not used. However, in our manuscript, the definition of “DFT” has been used more than once, which has been highlighted on Page 14, Page 15 and Page 18. So, it was very necessary to define “Density functional theory” in our manuscript.

Comment 3. Introduction: Hydrogen generation from water splitting is an intriguing research topic, and various strategies have been explored, including photocatalytic HER, photoelectrochemically catalytic HER, and electrocatalytic HER. Therefore, a brief survey of recent literature closely related to these strategies is highly desired to enrich the research background, for example, g-C₃N₄-based nanocomposites (Energy Environ. Sci. 2019, 12, 2080-2147) for photocatalytic HER; Ag-Based nanocomposites (Nanoscale, 2019, 11, 7062-7096) for electrocatalytic and photocatalytic HER, etc.

Response 3. We appreciate the reviewer very much for the helpful suggestion. Following your suggestion, we have added a brief introduction of recent literature closely related to these strategies for the HER catalysts. Besides, the recommended references have been cited in the appropriate places.

Accordingly, we have added a sentence on Page 3 in the revised manuscript to address this comment as follows:

“Recently, various strategies have been developed to realize the highly efficient catalysis for hydrogen evolution reaction (HER), including semiconductor-based photocatalytic HER (polymer g-C₃N₄,³ Ag/semiconductor,⁴ etc.), photoelectrochemically catalytic HER,⁵ and metal based electrocatalytic HER (metal sulfides,^{1,6,7} metal carbides,^{8,9} etc.).”

Additionally, we have cited Ref. 1, Ref. 3, Ref. 4, Ref. 6, Ref. 7, Ref. 8 and Ref. 9 in the appropriate places of the sentence.

Comment 4. Figure 3b, title for X axis: Please change “wavelenth” to “wavelength”.

Response 4. We appreciate the reviewer very much for the helpful suggestion. We have changed the “wavelenth” to “wavelength” in Figure 3b, which has been highlighted on Page 7 in the revised manuscript.

The above-mentioned Figure 3 has been revised as follows:

Fig. 3 Phase transition from 1T to 2H. **a**, Raman spectra of V₂O₃ film (black), V SACs@1T-WS₂/V₂O₃ (orange) and 2H-WS₂ (blue); **b**, PL spectra of V₂O₃ film (black), V SACs@1T-WS₂/V₂O₃ (orange) and 2H-WS₂ (blue); **c**, High-resolution XPS spectra of W 4f (left) and S 2p (right) core level peak regions for 2H and 1T phase, respectively. The fitting blue and pink curves represent the contributions of 1T and 2H phases, respectively; **d**, Schematic representation of the 1T to 2H structural phase transition occurring upon thermal annealing; **e**, Raman spectra of 2H-WS₂, V SACs@1T-WS₂, and V SACs@1T-WS₂ with different annealing temperatures in H₂/Ar for 2h.

Comment 5. Supplementary Figure 1, caption: please delete the description “The blue and yellow balls represent ...” because these balls and their meanings have been illustrated in the schematic models.

Response 5. We appreciate the reviewer very much for the helpful suggestion. We agree with the reviewer that the description of “the blue and yellow balls represent...”

was not necessary. Hence, we have deleted the description in the Supplementary Fig. 1, which has been highlighted in the revised supplementary information.

Comment 6. *Supplementary Fig. 10, caption: a) please change “different grown temperature” to “different grown temperatures”, b) please change “was probably reaction” to “will probably react”. Besides, please also change “x” to italic in the equation 1 to avoid any confusion.*

Response 6. We appreciate the reviewer very much for these helpful suggestions. Following your suggestions, we have changed “different grown temperature” to “different grown temperatures” and “was probably reaction” to “will probably react”. Besides, the “*x*” in the equation 1 has been changed to italic to avoid any confusion. All the revision has been highlighted in Supplementary Fig. 11.

Please note that the above-mentioned Supplementary Fig. 10 in the original supplementary information has been revised as Supplementary Fig. 11 in the revised supplementary information.

Comment 7. *Supplementary Fig. 12: Please change “fresh prepared” to “freshly prepared”*

Response 7. We appreciate the reviewer very much for the helpful suggestion. Following your suggestion, we have changed “fresh prepared” to “freshly prepared”, which has been highlighted in Supplementary Fig. 13 in the revised supplementary information.

Please note that the above-mentioned Supplementary Fig. 12 in the original supplementary information has been revised as Supplementary Fig. 13 in the revised supplementary information. The revised Supplementary Fig. 13 was displayed as follows:

Supplementary Fig. 13 Raman spectra of freshly prepared V SACs@1T-WS₂ sample on the V₂O₃ film (black plot) and after keeping for one year (red plot).

Comment 8. *Supplementary Fig. 13, Caption: The authors described “Please note that the 1T sample was annealed at different temperatures in H₂/Ar for 2h”. Does it mean that the same 1T sample was subjected to a series of heat treatments at different temperatures or three 1T samples were subjected to heat treatments at different temperatures? In the later case, please change “1T sample was” to “1T samples were”.*

Response 8. We appreciate the reviewer very much for the question. We are so sorry that we did not clearly describe the caption of Supplementary Fig. 13. In our experiments, three 1T samples were subjected to heat treatments at different temperatures for the thermal stability measurements. The caption has been changed to “**Supplementary Fig. 14** High resolution XPS spectra of W 4f (a) and S 2s (b) core level peak regions for 1T-200 °C, 1T-300 °C, 1T-400 °C samples. The fitting blue and pink curves represent the contributions of 1T and 2H phases, respectively. Please note that three pieces of 1T samples on the sapphire substrates were annealed at different temperatures in H₂/Ar for 2h” to avoid any confusion.

Please note that the above-mentioned Supplementary Fig. 13 in the original supplementary information has been revised as Supplementary Fig. 14 in the revised supplementary information. The revised Supplementary Fig. 14 was displayed as follows:

Supplementary Fig. 14 High resolution XPS spectra of W 4f (a) and S 2s (b) core level peak regions for 1T-200 °C, 1T-300 °C, 1T-400 °C samples. The fitting blue and pink curves represent the contributions of 1T and 2H phases, respectively. **Please note that three pieces of 1T samples on the sapphire substrates were annealed at different temperatures in H₂/Ar for 2h.**

Comment 9. *Supplementary Figure 14b: Please use the same color for the same sample shown in Supplementary Figure 14a to avoid any confusion. In addition, in the caption: please change “partially 1T phase transformation” to “partial 1T phase transformation”*

Response 9. We appreciate the reviewer very much for these helpful suggestions. Following your suggestions, same colors were used to represent the same samples in Supplementary Fig. 15 and the caption of “partially 1T phase transformation” has been changed to “partial 1T phase transformation”.

Please note that the above-mentioned Supplementary Fig. 14 in the original supplementary information has been revised as Supplementary Fig. 15 in the revised supplementary information. The revised Supplementary Fig. 15 was displayed as follows:

Supplementary Fig. 15 a, PL spectra of 2H-WS₂, V SACs@1T-WS₂, and V SACs@1T-WS₂ with different annealing temperatures in H₂/Ar for 2h; b, Enlarged PL spectra of V SACs@1T-WS₂ samples with different annealing temperatures. For all the V SACs@1T-WS₂ samples, the intensity of PL are almost completely suppressed, indicating the typical metallic behavior⁹. Meanwhile, compared with V SACs@1T-WS₂-200 °C sample and V SACs@1T-WS₂-300 °C sample, the PL intensity showed much higher intensity and blue shift in the V SACs@1T-WS₂-400 °C sample, which may be attributed to band-structure modification due to the partial 1T phase transformation to 2H phase.

Comment 10. *Supplementary Figure 15c-f: The values in X axis in Figure c show a decrease style from 1200 to 0, while the values in X axis in Figures e-f start from a lower one. Please make changes to make sure the values in X axis in these figures show a similar trend (decreasing or increasing).*

Response 10. We appreciate the reviewer very much for the helpful suggestion. Following your suggestion, we have changed the values in X axis in Supplementary Fig. 16c-f in a decreasing trend. Additionally, we have renamed the investigated V₂O₃ film

in the revised manuscript, which has been highlighted on Page 9-11. Therefore, we have made corresponding changes in the revised Supplementary Fig. 16.

Please note that the above-mentioned Supplementary Fig. 15 in the original supplementary information has been revised as Supplementary Fig. 16 in the revised supplementary information. The revised Supplementary Fig. 16 was displayed as follows:

Supplementary Fig. 16 a, Raman spectra of pure sapphire (black), V₂O₃ film (olive) and V₂O₃-Nuclei film (blue); **b**, XRD spectra of V₂O₃-Nuclei film (blue), V₂O₃-No Nuclei film (orange) and V₂O₃ film (olive); **c**, Survey spectra of V₂O₃- Nuclei film on sapphire substrate; **d-f**, High-resolution XPS spectra showing the S 2p (d), Al 2p (e) and V 2p (f) core levels for the V₂O₃-Nuclei film on the sapphire substrate sample. Please note that V₂O₃-No Nuclei film was prepared using VCl₃ precursor only. The oxygen was deduced from the residue oxygen in the tube furnace. The Raman spectrum of V₂O₃ film (olive) in Fig. S16a was measured on the 1T/V₂O₃ sample where no WS₂ appeared. The XRD pattern in Fig. S16b (olive) was obtained on the 1T/V₂O₃ sample.

Comment 11. *Supplementary Figure 16a-d: The images for the EDX mapping are little blurry. The authors are suggested to provide images with a higher resolution. In addition, scale bars are also suggested to provide for these images.*

Response 11. We appreciate the reviewer very much for these helpful suggestions. Following your suggestions, we have added higher resolution images for the EDX

mapping in Supplementary Fig. 17. The scale bars were also provided in the images. Please note that the above-mentioned Supplementary Fig. 16 in the original supplementary information has been revised as Supplementary Fig. 17 in the revised supplementary information. The revised Supplementary Fig. 17 was displayed as follows:

Supplementary Fig. 17 a, Elemental mapping of V_2O_3 -Nuclei film. The film was scratched from sapphire substrate for HRTEM measurement; **b**, EDX spectrum of V_2O_3 -Nuclei film. Scale bars: 40 nm.

Comment 12. *Supplementary Figure 17a-b: scale bars were provided but without any description(s).*

Response 12. We appreciate the reviewer very much for the helpful suggestion. Following your suggestion, we have added the scale bars for the images in Supplementary Fig. 18a-b, which has been highlighted in the revised supplementary information. Please note that the above-mentioned Supplementary Fig. 17 in the original supplementary information has been revised as Supplementary Fig. 18 in the revised supplementary information.

Comment 13. *Supplementary Figure 18a-d: scale bears were provided but without any description(s).*

Response 13. We appreciate the reviewer very much for the helpful suggestion. Following your suggestions, we have added the scale bars in Supplementary Fig. 19a-b. Besides, we have labeled the 1T- and 2H-WS₂ domains produced on the V₂O₃ film. Please note that the above-mentioned Supplementary Fig. 18 in the original supplementary information has been revised as Supplementary Fig. 19 in the revised supplementary information. The revised Supplementary Fig. 19 was displayed as follows:

Supplementary Fig. 19 a, Optical micrograph of 1T domains formed on the V₂O₃-Nuclei film; b, Optical micrograph of 2H domains formed on the V₂O₃-No Nuclei film; c, Optical micrograph of fresh commercial V₂O₃ film; d, Optical micrograph of used commercial V₂O₃ film; e, Raman spectra of 1T domains formed on the V₂O₃-Nuclei film (black plot) and 2H domains formed on the V₂O₃-No Nuclei film (red plot); f, PL spectrum of 2H domains on the V₂O₃-No Nuclei film; g, Raman spectra of commercial V₂O₃ film before (black plot) and after (red plot) CVD growth. Scale bars: a-b, 10 μm.

Comment 14. *Supplementary Figure 19b, caption: It would be better to change “Binding energy of 1T-VS₂/2H-WS₂ and 1T-VS₂/1T-WS₂” to Binding energy of 1T-VS₂/1T-WS₂ and 1T-VS₂/2H-WS₂”*

Response 14. We appreciate the reviewer very much for the helpful suggestion. Following your suggestion, we have changed the caption of “Binding energy of

1T-VS₂/2H-WS₂ and 1T-VS₂/1T-WS₂” to “1T-VS₂/1T-WS₂ and 1T-VS₂/2H-WS₂” in Supplementary Fig. 22. Please note that the above-mentioned Supplementary Fig. 19 in the original supplementary information has been revised as Supplementary Fig. 22 in the revised supplementary information.

Comment 15. *Supplementary Figure 20, caption: It would better to change “HER performance comparison with different mass loadings of catalysts” to “HER performance comparison for the various catalysts with different mass loadings (the catalysts include: ...”*

Response 15. We appreciate the reviewer very much for the helpful suggestion. Following your suggestion, we have changed the caption of “HER performance comparison with different mass loadings of catalysts” to “HER performance comparison for the various catalysts with different mass loadings (the catalysts include: ...”, which has been highlighted in Supplementary Fig. 34. Please note that the above-mentioned Supplementary Fig. 20 in the original supplementary information has been revised as Supplementary Fig. 34 in the revised supplementary information.

Comment 16. *Supplementary Figure 20, Y axis: please change “slop” to “slope”.*

Response 16. We appreciate the reviewer very much for the helpful suggestion. Following your suggestions, we have changed the Y axis of “slop” to “slope” in Supplementary Fig. 34. Please note that the above-mentioned Supplementary Fig. 20 in the original supplementary information has been revised as Supplementary Fig. 34 in the revised supplementary information. The revised Supplementary Fig. 34 was displayed as follows:

Supplementary Fig. 34 HER performance comparison for the various catalysts with different mass loadings (the catalysts include strained 1T'-WS₂,¹⁹ 1T'-MoS₂ nanosheets,²⁰ 1T'-MoS₂ monolayers,³ 1T' WSe₂,²¹ 2H-1T boundary MoS₂,²² vacancy sites-2H MoS₂,²³ 1T-MoS₂ nanosheets,²⁴ P-1T-MoS₂,²⁵ (N,PO₄³⁻)-1T-MoS₂,²⁶ SA Co-D-1T-MoS₂,²⁷ SA Ni-2H-MoS₂,²⁸ SA Rh-2H-MoS₂,²⁹ PE-CVD 1T-WS₂³⁰). The red dashed circles represent the 1T(1T')-W based TMDs.

Comment 17. *Supplementary Figure 21: what do they stand for by the different balls shown in the model structures?*

Response 17. We appreciate the reviewer very much for the helpful suggestion. Following your suggestion, we have added the description of different balls in the model structures in Supplementary Fig. 37. They represent H, S, W and V element, respectively, from up to down. Please note that the above-mentioned Supplementary Fig. 21 in the original supplementary information has been revised as Supplementary Fig. 37 in the revised supplementary information.

Comment 18. *Table S1: please change “Synthesized strategy” to “synthesis strategies”, and “Ref.” to “References”*

Response 18. We appreciate the reviewer very much for the helpful suggestion. Following your suggestion, we have changed “Synthesized strategy” to “synthesis strategies” and “Ref.” to “References” in the Table S1, which has been highlighted in the revised supplementary information.

Comment 19. *Table S4: please change “Ref.” to “Reference”*

Response 19. We appreciate the reviewer very much for the helpful suggestion. Following your suggestion, we have changed “Ref.” to “References” in the Table S7, which has been highlighted in the revised supplementary information. Please note that the above-mentioned Table S4 in the original supplementary information has been revised as Table S7 in the revised supplementary information.

Response to Reviewer #2

Comment. *Currently, it remains a huge challenge to directly synthesize 1T-WS₂, since it has the highest formation energy when compared with other transition metal dichalcogenides (TMDs) from Group VI elements with 1T phase. In this work, the authors have implemented a facile VCl₃-assist chemical vapor deposition (CVD) strategy to introduce atomically dispersed V-atoms onto 1T-WS₂ monolayers, which is a very original and heuristic work. Impressively, the authors have performed the detailedly experimental and theoretical investigations on the growth mechanism and HER catalytic activity of V SACs on 1T-WS₂ monolayers. I highly recommend this work to be published on the Nature Communications. However, the following questions should be addressed before its publication:*

Response. We appreciate the reviewer very much for the highly positive comment. His/Her concerns are addressed as follows.

Comment 1. *In this work, the authors have prepared the 1T-WS₂ monolayers with the help of VCl₃, as 2H-WS₂ monolayers were obtained without VCl₃. This point should be quite interesting. However, I am wondering if the amount of VCl₃ has a great impact on the purity of 1T-WS₂.*

Response 1. We appreciate the reviewer very much for the insightful question. We have made experiments to demonstrate the amount of VCl₃ has a great impact on the purity of 1T-WS₂. The WS₂ samples were prepared by using different amount of VCl₃ (from 0.01 g to 0.10 g). The corresponding optical micrographs and Raman spectra were shown in Supplementary Fig. 23-25. Apparently, when the amount of VCl₃ was

0.01 g, no WS₂ domains were formed on the sapphire substrate. As demonstrated in Supplementary Fig. 24 (black plot) and Supplementary Fig. 25 (black plot), the peaks that assigned to V₂O₃ were very weak, indicating the poor quality of V₂O₃ film.

In addition, when the amount of VCl₃ was increased to 0.02 g, dominated 2H-WS₂ samples were formed on the surface of V₂O₃ film. Particularly, the characteristic metallic peaks of J₁, J₂ and J₃ were gradually observed from Supplementary Fig. 24 (red plot). As demonstrated in the Fig. 4 in the revised manuscript, the 1T-VS₂ nuclei played important roles in determining the 1T-WS₂ phase growth. However, if the amount of VCl₃ was insufficient, it was deduced that only limited 1T-VS₂ nuclei were formed on the surface of V₂O₃ film to trigger the 1T-WS₂ growth. As a result, a low 1T/2H ratio of WS₂ were formed according to the XPS analysis (Supplementary Fig. 26).

Prominent of J₁, J₂ and J₃ peaks were clearly observed until the amount of VCl₃ was increased to 0.03 g, as shown in Supplementary Fig. 24 (blue plot). Simultaneously, high quality of V₂O₃ film was achieved if the VCl₃ amount was 0.03 g, as shown in Supplementary Fig. 25 (blue plot) and Supplementary Fig. 10 in the revised supplementary information. Appropriate formation of VS₂ nuclei and high quality of V₂O₃ were in favor of the epitaxial growth of 1T-WS₂ monolayers with high quality (Fig. 3c, 91% 1T-WS₂).

However, it was found than if the amount of VCl₃ was in the range of 0.05 g~0.08 g, no typical metallic peaks were observed from the Raman spectra in Supplementary Fig. 24 (olive and purple plots). Only 2H-WS₂ domains were formed, which were further corroborated by the XPS analyses (Supplementary Fig. 26). Moreover, it was believed that the large amount of VCl₃ could result in the high density of VO_x vapor phase during the heating process, which could significantly accelerate the deposition rate of V₂O₃ film, leading to a V₂O₃ film with rough surface. Particularly, poor quality of V₂O₃ films were achieved when the amount of VCl₃ was in the range of 0.05g~0.08 g, as shown in Supplementary Fig. 25 (olive and purple plots). Moreover, no WS₂ domains were formed if the amount of VCl₃ was ~0.1 g due to the extremely rough surface of V₂O₃ film. From the analysis of the influence of

amount of VCl_3 on the purity of 1T- WS_2 , we can conclude that the quality of V_2O_3 film was the constraining factor to determine the formation of 1T- WS_2 with high phase purity when the amount of VCl_3 was higher than 0.03 g. Whereas, the density of VS_2 nuclei was the constraining factor to impact the 1T- WS_2 formation when the amount of VCl_3 was less than 0.03 g.

Accordingly, the influence of amount of VCl_3 on the 1T- WS_2 growth has been added in the revised manuscript to address this question, which has been highlighted on Page 11 as follows:

“Additionally, it was also demonstrated that the amount of VCl_3 could significantly affect the controllable phase growth of WS_2 (see details in Figure S23-S26).”

The above-mentioned Supplementary Fig. 23-26 has been added in the revised supporting information as follows:

Supplementary Fig. 23 Optical micrographs of as-prepared products using different amount of VCl_3 . **a**, 0.01g; **b**, 0.02 g; **c**, 0.03 g; **d**, 0.05 g; **e**, 0.08 g; **f**, 0.10 g. $T_1=860\text{ }^\circ\text{C}$, $T_2 = 160\text{ }^\circ\text{C}$, $\text{Ar}/\text{H}_2 = 80\text{ sccm}/20\text{ sccm}$, $t = 15\text{ min}$; T_1 refers to the heating temperature of the furnace, T_2 refers to the heating temperature of the sulfur powder, and t refers to the growth time. All the other experimental parameters are the same. Scale bars: 5 μm .

Supplementary Fig. 24 The Raman spectra of as-prepared products using different amount of VCl_3 .

Supplementary Fig. 25 The Raman spectra of as-prepared V_2O_3 film using different amount of VCl_3 . The Raman spectra were measured on the areas where no WS_2 domains appeared.

Supplementary Fig. 26 High resolution XPS spectra of W 4f (a) and S 2s (b) core level peak regions for WS₂ domains using different amount of VCl₃. The fitting blue and pink curves represent the contributions of 1T and 2H phases, respectively. Please note that obtained WS₂ domains were transferred on the sapphire substrates for the XPS measurements.

The discussion of the effect of amount of VCl₃ on the 1T-WS₂ growth has been added in the revised supplementary information as follows:

“In order to demonstrate the amount of VCl₃ has a great impact on the phase purity of WS₂, we have prepared WS₂ samples using different mass amount of VCl₃ (from 0.01 g to 0.10 g). Whereas, the density of VS₂ nuclei was the constraining factor to impact the 1T-WS₂ formation when the amount of VCl₃ was less than 0.03 g.”

Comment 2. *Is the substrate of c-plane sapphire play an import role in the controllable phase growth? It is well-known that 2H-WS₂ monolayers are also able to epitaxially grow on the other substrate, such as SiO₂/Si wafer. Hence, if the authors use the same growth conditions, is it possible that the 1T phase samples are formed on the SiO₂/Si wafer or other kinds of substrates? The authors need to provide more information about the choice of substrate.*

Response 2. We appreciate the reviewer very much for these insightful questions. We have made more experiments to demonstrate that the c-plane sapphire also played an import role in the controllable phase growth. **Firstly**, we have tried to grow WS₂ samples on the SiO₂/Si wafer under the same growth condition, however, 2H-WS₂ domains were directly grown on the surface without formation of V₂O₃ film, as shown

in Figure R1. **Secondly**, the choice of the c-plane sapphire substrates was due to the similar corundum structure to the V_2O_3 film, which could promote the epitaxial growth of V_2O_3 film on the surface. As far as we know, both a-plane (110) and c-plane (001) sapphire substrates could promote the epitaxial growth of V_2O_3 film on the surface due to small in-plane lattice misfits (Table R1). The XRD patterns of a- and c-plane sapphire substrates were shown in Figure R2. The WS_2 growth on the a-plane sapphire substrate was performed under the same condition with c-plane sapphire. Surprisingly, no WS_2 domains were formed on the a-plane sapphire substrate, as demonstrated in Figure R3. In particular, the surface of V_2O_3 film on the a-plane sapphire substrate was extremely rough from the optical micrograph in Fig. R3a, which was probably caused by the three-dimensional (3D) Volmer-Weber growth mode of V_2O_3 film. (*J. Mater. Res.*, 2007, 22, 2825.) In contrast, V_2O_3 film growth mode on the c-plane sapphire substrate was layer-by-layer (Frank-van der Merwe) mode (*J. Mater. Res.*, 2007, 22, 2825.), leading to smooth surface of V_2O_3 film, which could significantly facilitate the epitaxial growth of 1T- WS_2 .

Figure R1. Optical micrograph (a) and Raman spectrum (b) of 2H- WS_2 domains grown on the SiO_2/Si wafer substrate. Scale bar: 5 μm .

Figure R2. XRD patterns of a-plane (red plot) and c-plane (black plot) of sapphire substrates.

Figure R3. Optical micrograph (a) and Raman spectrum (b) of V_2O_5 film grown on the a-plane sapphire substrate. $T_1=860 \text{ }^\circ\text{C}$, $T_2 = 160 \text{ }^\circ\text{C}$, $\text{Ar}/\text{H}_2 = 80 \text{ sccm}/20 \text{ sccm}$, $t = 15 \text{ min}$; T_1 refers to the heating temperature of the furnace, T_2 refers to the heating temperature of the sulfur powder, and t refers to the growth time. The amount of VCl_3 and WO_3 was 0.03 g and 0.30 g , respectively. Scale bar: $5 \mu\text{m}$.

Table R1. Theoretical in-plane lattice misfits of V₂O₃ film on the sapphire substrate.

	Lattice parameter (nm)	
	V ₂ O ₃	Al ₂ O ₃
a	0.4958	0.4758
c	1.3958	1.2992

Comment 3. *According to the author's statement, the V SACs on the 1T-WS₂ monolayers showed stable HER performance, however, there was no convincing evidence to exhibit the stability of the structure after HER test. Here, it is suggested that the authors add some tests, such as Raman, STEM, to show the structure after catalysis.*

Response 3. We appreciate the reviewer very much for these helpful suggestions. Following your suggestions, Raman spectroscopy and STEM imaging were performed to demonstrate the stability of V SACs 1T-WS₂ electrocatalyst after HER test. As shown in Supplementary Fig. 36a, obvious metallic peaks (J₁, J₂, J₃) of V SACs 1T-WS₂ electrocatalyst after HER test were clearly observed (red plot). Moreover, the typical 1T structure of WS₂ after stability test was confirmed according to the STEM image in Supplementary Fig. 36b.

Accordingly, the description of metallic properties of V SACs 1T-WS₂ after HER test has been added in the revised manuscript to address this question, which has been highlighted on Page 13 as follows:

“The metallic properties of V SACs 1T-WS₂ catalyst after stability test were also investigated by Raman spectroscopy (Figure S36a), which showed obvious metallic peaks (J₁, J₂, J₃) in the Raman spectrum (red plot). Moreover, the STEM image showed in Figure S36b confirmed the V SACs 1T-WS₂ structure after stability test. Both the Raman spectrum and STEM image indicated the robust 1T structure of V SACs 1T-WS₂ catalyst after HER test.”

The above-mentioned Supplementary Fig. 36 has been added in the revised supporting information as follows:

Supplementary Fig. 36 Raman spectra (a) and STEM image (b) of V SACs 1T-WS₂ catalyst after stability test. The V SACs 1T-WS₂ catalyst after stability test was transferred on the sapphire substrate and TEM grid for the Raman spectrum and STEM measurements, respectively. The white dashed circles represent the V SACs. Scale bar: 1 nm.

Comment 4. *A technical question: in the AFM image of Supplementary Fig. 7a for demonstrating the thickness of WS₂ monolayer, why the topographic image was so blurry? Hence, it was very hard to clearly identify the morphology of WS₂.*

Response 4. We appreciate the reviewer very much for the question. The blurry topographic image was caused by the small height color contrast between V₂O₃ film and 1T-WS₂ monolayer, which has been marked by the black rectangle in Fig. R4a. To clearly distinguish the morphology of WS₂, the corresponding phase image of WS₂ on the V₂O₃ surface was acquired, as shown in Fig. R4b, indicating the circular morphology of 1T-WS₂.

Figure R4. Topographic image (a) and phase image (b) of 1T-WS₂ monolayer grown on the V₂O₃/sapphire substrate. Scale bars: 5 μ m.

Comment 5. Although the author has cited the paper for the calculations of TOFs, please specify the details for the TOFs in this work.

Response 5. We appreciate the reviewer very much for the helpful suggestion. Following your suggestion, the details for the TOF were specified in the revised manuscript to address this suggestion, which has been highlighted on Page 19 as follows:

“Calculation of Turnover Frequency

The TOF calculation details were specified as below, which was reported elsewhere.

$$\text{TOF} = \frac{\text{Total hydrogen turnovers per geometric area}}{\text{active sites per geometric area}}$$

The total hydrogen turnovers were calculated from the current density in the LSV polarization curve according to the equation as below:

$$\begin{aligned} & \text{Total hydrogen turnovers} \\ &= (|j| \frac{\text{mA}}{\text{cm}^2}) \left(\frac{1\text{C/s}}{1000 \text{ mA}} \right) \left(\frac{1 \text{ mol } e^-}{96485 \text{ C}} \right) \left(\frac{1 \text{ mol}}{2 \text{ mol } e^-} \right) \frac{(6.022 \times 10^{23} \text{ moleculars } H_2)}{1 \text{ mol } H_2} \end{aligned}$$

The number of active sites in the V SACs@1T-WS₂ catalyst was obtained from the mass loading on the glass carbon electrode.

Active sites

$$= \left(\frac{\text{electrocatalyst loading per geometric area} \left(\frac{\text{g}}{\text{cm}^2} \right) \times \text{Vwt}\%}{V M_w \left(\frac{\text{g}}{\text{mol}} \right)} \right) \left(\frac{6.022 \times 10^{23} \text{V atoms}}{1 \text{ mol V}} \right)$$

”

Comment 6. The catalytic tests to demonstrate the stability of the catalysts are incomplete. Evidently, the authors have shown the stability of catalysts after continuous CV cycles. However, this is not enough. Please carry out the stability test of long-term electrolysis (at least 10 h).

Response 6. We appreciate the reviewer very much for the helpful suggestion. Following your suggestion, the stability test was conducted at current densities higher than 10 mA/cm² in 0.5 M H₂SO₄ electrolyte for 100 h. As revealed by the chronoamperometric curve of V SACs 1T-WS₂ electrocatalyst in Supplementary Fig. 35, the current density for the V SACs 1T-WS₂ electrocatalyst displayed a slight

current decay of 1.0 mA cm^{-2} after 24 h and 3.4 mA cm^{-2} after 100 h, indicating a high stability of V SACs 1T-WS₂ catalyst.

Accordingly, the description of the electrochemical stability test has been added in the revised manuscript to address this comment, which has been highlighted on Page 13 as follows:

“The stability test was conducted at current densities higher than 10 mA/cm^2 in $0.5 \text{ M H}_2\text{SO}_4$ electrolyte for 100 h. As revealed by the chronoamperometric curve of V SACs 1T-WS₂ electrocatalyst in Supplementary Fig. 35, the current density for the V SACs 1T-WS₂ electrocatalyst displayed a slight current decay of 1.0 mA cm^{-2} after 24 h and 3.4 mA cm^{-2} after 100 h, indicating a high stability of V SACs 1T-WS₂ catalyst.”

The above-mentioned Supplementary Fig. 35 has been added in the revised supporting information as follows:

Supplementary Fig. 35 Chronoamperometric curve of V SACs 1T-WS₂ catalyst at an overpotential of 400 mV in $0.5 \text{ M H}_2\text{SO}_4$ electrolyte.

Response to Reviewer #3

Comment: *In this work, the authors demonstrated a one-step CVD growth of V-doped 1T-WS₂. The authors also proposed a 1T-VS₂ nucleation and templated*

growth model, and discussed the importance of V_2O_3 substrate for 1T- WS_2 growth. Furthermore, HER catalysis properties of the V SACs@1T- WS_2 was tested, exhibiting enhanced HER properties compared to undoped 1T- WS_2 . This is an interesting piece of work containing both experimental and theoretical efforts. However, my main concern is the effects of V_2O_3 . It seems the formation of V_2O_3 underneath the material of interest is unavoidable, and it can have influence on different characterization techniques (e.g. EELS, XPS, HER properties measurement). It is also unclear to me whether the V_2O_3 is removed during transfer. In addition, considering the fact that V compounds can be easily oxidized when exposing to air, and the authors did not mention that all the characterizations were carried out without exposing them to air, thus the characterizations and proposed mechanisms seem not reliable. The manuscript is also hard to read in some sections. Thus, I would recommend a rejection of the current manuscript, but if the authors address all the points, the paper could be eventually published.

Response: We are pleased that the reviewer commented our work as “interesting” piece of work containing both experimental and theoretical efforts. His/Her concerns are addressed as follows.

Details are described below:

Comment 1. *In Fig. 2f and Line 116-118 of the main manuscript, EELS was performed to identify V dopants. However, since there is a V_2O_3 film underneath the V SACs@1T- WS_2 , signals of V can also from the V_2O_3 . Can the authors comment on how to rule out this possibility?*

Response 1. We appreciate the reviewer very much for the insightful question. EELS was a very powerful and sensitive analytical technique to determine the local coordination state of V single atoms. To identify the V dopants in the 1T- WS_2 monolayers, we have transferred the V SACs@1T- WS_2 monolayers on the TEM grid for the STEM image and EELS analysis. We have explicitly described the transferring process in both the original manuscript and revised manuscript on Page 16 as follows:

“The as-grown samples were transferred onto arbitrary substrates, such as fresh

sapphire, SiO₂/Si, glass carbon, and holy-carbon nickel TEM Grid using a modified method in our lab.”

In particular, the EELS of V SACs@1T-WS₂ in Fig. 2e showed two major features of V peaks assigned to V⁴⁺, indicating the V substitutions in the 1T-WS₂ layer. However, the valence state of V in the V₂O₃ was +3, which was different with the V valence state in the 1T-WS₂ layer. In conclusion, the signals of V EELS from V₂O₃ could be ruled out.

Comment 2. *Was any image filtering process used for enhancing the visibility of atoms in Fig. 2c and d? This should be included in experimental details.*

Response 2. We appreciate the reviewer very much for the helpful suggestion. In our STEM image process, radial wiener filter was processed to enhance the visibility of atoms in Fig. 2c-2d. Following your suggestion, we have added the filtering process in the revised manuscript to address this suggestion, which has been highlighted on Page 17 as follows:

“Radial wiener filter was carried out to enhance the visibility of atoms.”

Comment 3. *In Line 155-157 of the manuscript, please check the resolution of the spectrometer to make sure all digits are significant. For example, “418.44” might be changed to “418.4”.*

Response 3. We appreciate the reviewer very much for these helpful suggestions. Following your suggestions, we have carefully checked the resolution of the spectrometer. Accordingly, we have changed the Raman peaks at “418.44 cm⁻¹” to “418.4 cm⁻¹”, “354.60 cm⁻¹” to “354.6 cm⁻¹”, “147.85 cm⁻¹” to “147.9 cm⁻¹”, “214.50 cm⁻¹” to “214.5 cm⁻¹” and “385.32 cm⁻¹” to “385.3 cm⁻¹”, which have been highlighted on Page 7-8 in the revised manuscript.

Comment 4. *For the phase purity analysis (Fig. 3c), if I understand correctly, the XPS comparison was done between pure 2H-WS₂ and V SACs@1T-WS₂/V₂O₃. I think the XPS peak shift can also originate from the interaction with V₂O₃, and also*

the effect of V doping, which can not be ruled out in this case, and makes the phase purity analysis not very convincing.

Response 4. We appreciate the reviewer very much for the question. XPS was a very powerful technique to detect the oxidation state and coordination geometry of W element. To quantitatively identify the phase purity of 1T monolayers, we have transferred the V SACs@1T-WS₂ monolayers on the sapphire substrate for the XPS analyses.

To make more clearer for the XPS analysis, we have added one sentence on Page 8 to address this comment. The added sentence was described as follows:

“To illustrate the high purity of the obtained 1T phase, XPS spectra were performed to quantify the 1T and 2H compositions according to the high sensitivity of tungsten signal to its oxidation state and coordination geometry.^{1,47} The V@SACs@1T-WS₂ monolayers were transferred on the fresh sapphire substrate for the XPS measurements.”

We have explicitly described the transferring process in both the original manuscript and revised manuscript on Page 16 as follows:

“The as-grown samples were transferred onto arbitrary substrates, such as fresh sapphire, SiO₂/Si, glass carbon, and holy-carbon nickel TEM Grid using a modified method in our lab.”

We have also explicitly described the details of XPS measurements in both the original manuscript and revised manuscript on Page 17 as follows:

“For the XPS and Raman spectra of V SACs@1T-WS₂ annealed at different temperatures, the V SACs@1T-WS₂ monolayers were transferred on the fresh sapphire substrates and annealed in H₂/Ar condition with different temperatures.”

As demonstrated by the EELS, no V₂O₃ contamination was involved in the transferred 1T monolayers, as a result, the possibility of XPS shift caused by the V₂O₃ film could be ruled out. Additionally, no V signals of high-resolution V 2p XPS were detected in the transferred 1T monolayers due to the detection limit of XPS (Figure 5R). Since the V signals were not detected, its effect on the XPS peak shift was not significant. Through a series of careful analyses, our phase purity analysis of 1T-WS₂

monolayers by the XPS spectra was very convincing.

Figure R5. High-resolution of V 2p in the transferred 1T-WS₂ monolayers on the sapphire substrate.

Comment 5. *It seems the HER measurement was done on the V SACs@1T-WS₂/V₂O₃. Thus, similar to the previous question, could V₂O₃ contribute and enhance the measured catalytical performance?*

Response 5. We appreciate the reviewer very much for the insightful question. In our HER measurement of V SACs@1T-WS₂ monolayers, however, it was performed on a glass carbon electrode due to the insulating sapphire substrate. In particular, the V SACs@1T-WS₂ electrocatalyst was transferred on the glass carbon electrode.

We have explicitly described the transferring process in both the original manuscript and revised manuscript on Page 16 as follows:

“The as-grown samples were transferred onto arbitrary substrates, such as fresh sapphire, SiO₂/Si, glass carbon, and holy-carbon nickel TEM Grid using a modified method in our lab.”

We have also explicitly described the details of HER measurement in both the original manuscript and revised manuscript on Page 11 as follows:

“The as-produced V SACs@1T-WS₂ monolayers were transferred on the glass carbon (GC) electrode for the HER performance measurement using a three-electrode setup in 0.5 M H₂SO₄.”

As demonstrated by the EELS analysis in Fig. 2e, no V_2O_3 film was involved in the transferred 1T monolayers. Moreover, the HER performance was carried out in the harsh acid electrolyte (0.5 M H_2SO_4), which could etch the V_2O_3 film if the V_2O_3 was used as the electrocatalyst. Consequently, the possibility of enhanced HER performance caused by the V_2O_3 film could be ruled out.

Comment 6. *Figure resolution need to be improved, and the label font size need to be the same.*

Response 6. We appreciate the reviewer very much for these helpful suggestions. Following your suggestions, we have carefully checked all the Figs. and made corresponding revision, which have been highlighted in the revised manuscript and supplementary information.

The resolution has been enhanced in the revised Supplementary Fig. 8, Supplementary Fig. 9 and Supplementary Fig. 17.

The label font size has been kept the same in the revised Supplementary Fig. 6, Supplementary Fig. 7, Supplementary Fig. 16 and Supplementary Fig. 18.

Particularly, Supplementary Fig. 8 in the original supplementary information has been divided into Supplementary Fig. 8 and Supplementary Fig. 9 in the revised supplementary information to enhance the resolution of the images. The revised Figs. were displayed as follows:

Supplementary Fig. 6 **a**, Raman spectra of 1T-V₂O₃ region with different annealing temperatures. 1T features in Raman spectra get relatively stable signals with the annealing temperature range from 25 °C to 300 °C in air; **b**, Raman spectra of V₂O₃ region with different annealing temperatures. The V₂O₃ film becomes VO₂ (M) at 400 °C in air for 5 mins; **c-d**, Optical micrographs of as-grown 1T-V₂O₃ phase on V₂O₃ film after annealing under different temperatures for 5 mins. 300 °C (c) and 400 °C (d). Scale bars: c-d, 10 μm.

Supplementary Fig. 7 AFM imaging of pristine and transferred 1T-WS₂. **a**, topographic image of pristine V SACs@1T-WS₂ on V₂O₃/sapphire; **b**, pristine height profile; **c**, topographic image of transferred V SACs@1T-WS₂ on sapphire; **d**, Transferred height profile. Scale bars: a, 5 μm; c, 5 μm.

Supplementary Fig. 8 **Optical characterization of 1T and 2H phase.** Optical micrograph and fluorescence of V SACs@1T-WS₂ monolayer, 2H-WS₂ monolayer and trapezoidal 1T/2H bilayer heterostructure (formed by stacking of a 1T monolayer and a 2H monolayer). Fluorescence image was taken with a color camera, showing a dark emission state of V SACs@1T-WS₂, and bright emission state of 2H-WS₂ and 1T/2H bilayer heterostructure junction. The exposure time was 2000 ms.

Supplementary Fig. 9 Optical characterization of 1T and 2H phase. **a**, SHG intensity collected from V SACs@1T-WS₂, 2H-WS₂ and bilayer 1T/2H heterostructure samples on SiO₂/Si substrates, respectively; **b**, Raman spectra of monolayer V SACs@1T-WS₂ as a function of rotating angle; **c**, Polarization Raman intensity from V SACs@1T-WS₂ as a function of rotating angle. The polar pattern of the A_{1g} mode in the 1T phase was well fitted by a circle (orange solid line), which showed isotropic Raman scattering.

Supplementary Fig. 16 **a**, Raman spectra of pure sapphire (black), V_2O_3 film (olive) and V_2O_3 -Nuclei film (blue); **b**, XRD spectra of V_2O_3 -Nuclei film (blue), V_2O_3 -No Nuclei film (orange) and V_2O_3 film (olive); **c**, Survey spectra of V_2O_3 -Nuclei film on sapphire substrate; **d-f**, High-resolution XPS spectra showing the S 2p (d), Al 2p (e) and V 2p (f) core levels for the V_2O_3 -Nuclei film on the sapphire substrate sample. Please note that V_2O_3 -No Nuclei film was prepared using VCl_3 precursor only. The oxygen was deduced from the residue oxygen in the tube furnace. The Raman spectrum of V_2O_3 film (olive) in Fig. S16a was measured on the 1T/ V_2O_3 sample where no WS_2 appeared. The XRD pattern in Fig. S16b (olive) was obtained on the 1T/ V_2O_3 sample.

Supplementary Fig. 17 **a**, Elemental mapping of V_2O_3 -Nuclei film. The film was scratched from sapphire substrate for HRTEM measurement; **b**, EDX spectrum of V_2O_3 -Nuclei film. Scale bars: 40 nm.

Supplementary Fig. 18 a-b, Optical micrographs of V_2O_3 -Nuclei film in the upstream (a) and downstream (b); **c**, Raman spectrum of V_2O_3 -Nuclei film in the downstream. The peak at 406.3 cm^{-1} was assigned to VS_2 and different with the peak at 418.3 cm^{-1} , which was derived from the A_{1g} mode of the substrate sapphire⁵. The recorded Raman peaks of VS_2 in Figure S18 were very weak, indicating that the amount of VS_2 was in very low concentration relative to V_2O_3 . **Scale bars: a-b, $90\text{ }\mu\text{m}$.**

Comment 7. *Could the authors provide the V 2p XPS? Did the authors consider the V-S bond when fitting the V 2p peak?*

Response 1. We appreciate the reviewer very much for these questions. Following your suggestion, we have provided the V 2p XPS spectrum in the transferred 1T- WS_2 monolayers, which was shown in Figure 5R. Apparently, no V signals were detected due to the detection limit of XPS. And we could not gain any useful information. However, the V signals of EELS was pronounced due to the relatively higher detection limit than XPS (Fig. 2e) and two major features of L peaks were assigned to V^{4+} , affirming the V-S bond in the 1T- WS_2 layer.

Figure R5. High-resolution of V 2p in the transferred 1T-WS₂ monolayers on the sapphire substrate.

Comment 8. *In Figure S18, the 1T or 2H WS₂ domains are not clear. Could the authors label them? Raman mapping is recommended, so that the WS₂ region can be clearly separated out. In figure s18c, we could also see some domain with different contrast, so it would be better to use Raman mapping to identify the phase and the phase distribution.*

Response 8. We appreciate the reviewer very much for these insightful suggestions. Following your suggestion, **firstly**, we have labeled the 1T or 2H domains in Supplementary Fig. 19.

Please note that Supplementary Fig. 18. in the original supplementary information was revised as Supplementary Fig. 19 in the revised supplementary information, which was displayed as follows:

Supplementary Fig. 19 a, Optical micrograph of 1T domains formed on the V₂O₃-Nuclei film; b, Optical micrograph of 2H domains formed on the V₂O₃-No Nuclei film; c, Optical micrographs of fresh commercial V₂O₃ film; d, Optical micrograph of used commercial V₂O₃ film; e, Raman spectra of 1T domains formed on the V₂O₃-Nuclei film (black plot) and 2H domains formed on the V₂O₃-No Nuclei film (red plot); f, PL spectrum of 2H domains on the V₂O₃-No Nuclei film; g, Raman spectra of commercial V₂O₃ film before (black plot) and after (red plot) CVD growth. Scale bars: a-b, 10 μm.

Secondly, we have made Raman mapping to identify the 1T or 2H domains formed on the different V₂O₃ films, as shown in Supplementary Fig. 20. Raman mapping was directly conducted on the WS₂@V₂O₃ film to identify the 1T or 2H domains formed on the different V₂O₃ films. Fig. S20a-b and Fig. S20c-d are taken on 1T and 2H domains, respectively. Obviously, homogeneous signals from J₁ mode of 1T character were detected all over the domain (Fig. S20a), indicating the WS₂ domains grown on the V₂O₃-Nuclei film were metallic 1T phase. The corresponding Raman spectrum of point A was displayed in Supplementary Fig. 21 (black plot), showing clear J₁ peak which only belongs to 1T metallic phase. The signals of E_{2g}¹ resonance modes of WS₂ were observed on both the domains formed on V₂O₃-Nuclei film (Supplementary Fig. 20b) and V₂O₃-No Nuclei film (Supplementary Fig. 20d), respectively, confirming the formation of WS₂. However, no signals of J₁ mode were detected in the WS₂ domain formed on the V₂O₃-No Nuclei film, indicating that the WS₂ domain was 2H semiconducting phase. The typical Raman spectrum taken at

point B in Supplementary Fig. 21 (red plot) clearly demonstrated the absence of J_1 peak, further confirming the 2H phase.

Moreover, the Raman mapping on fresh commercial V_2O_3 film was also carried out to identify the domains with different contrast. Apparently, no signals of E_{2g}^1 resonance modes of WS_2 were detected in Supplementary Fig. 20e, indicating no WS_2 was formed on the surface. However, the signals of A_{1g} mode of V_2O_3 film was observed, demonstrating the V_2O_3 structure.

Accordingly, the description of 1T or 2H domains confirmed by the Raman mapping was added in the revised manuscript to address this comment, which has been highlighted on Page 11 as follows:

“The 1T- or 2H- WS_2 domains were also confirmed by the Raman mapping in Figure S20-S21.”

Additionally, the Raman mapping analyses were added in the revised supplementary information as follows:

“Raman mapping was directly conducted on the $WS_2@V_2O_3$ film to identify the 1T or 2H domains formed on the different V_2O_3 films, as shown in Supplementary Fig. 20...However, the signals of A_{1g} mode of V_2O_3 film was observed, demonstrating the V_2O_3 structure.”

The above-mentioned Supplementary Fig. 20-21 have been added into the revised supplementary information as follows:

Supplementary Fig. 20 a-b, Raman mapping images of a 1T-WS₂ flake obtained in the J₁ (a) and E_{2g}¹ (b) vibrational modes, respectively. The measured WS₂ flake was grown on the V₂O₃-Nuclei film substrate; **c-d**, Raman mapping images of a 2H-WS₂ flake obtained in the J₁ (c) and E_{2g}¹ (d) vibrational modes, respectively. The measured WS₂ flake was grown on the V₂O₃-No Nuclei film substrate; **e-f**, Raman mapping image of the fresh commercial V₂O₃ film obtained in the E_{2g}¹ (WS₂) vibrational mode (e) and A_{1g} (V₂O₃) mode. Scale bars: a-d, 1 μm; e-f, 5 μm.

Supplementary Fig. 21 Raman spectra of the selected three points (A, B, C) on their respective substrates in Supplementary Fig. 20a, 20d and 20f, respectively.

Finally, to further confirm the compositions of the domains with different contrast in Supplementary Fig. 19c, scanning electron microscopy (SEM) and energy dispersive spectroscopy (EDS) were conducted on the fresh commercial V₂O₃ film, as shown in Figure R6-R8. As indicated by Fig. R6, five areas were marked for the EDS elemental analyses. From the EDS spectra, V, O, Al and C elements were detected. V was from the V₂O₃ film while Al was from the sapphire substrate. O was from the V₂O₃ film and the sapphire substrate. C was from the conducting coating layer, which was used to enhance the conductivity of sample. Particularly, the EDS spectra of spot 2, spot 3 and spot 4 were selected on the particles on the surface. From the compositions in the spectra, the observed particles were mainly V₂O₃, which were marked by the white arrows in Supplementary Fig. 19c. Importantly, no W and S elements were detected on the selected five areas, indicating the pure V₂O₃ composition of fresh commercial V₂O₃ film.

Figure R6. a, SEM image of fresh commercial V_2O_3 film on the sapphire substrate. **b**, EDS spectrum for the full area 1. Scale bar: 2 μm .

Figure R7. EDS spectrum for the spot 2 (a) and spot 3 (b).

Figure R8. EDS spectrum for the spot 4 (a) and spot 5 (b).

Comment 9. *The sample labeling may be a bit confusing. For example, from line 351 to 352: “To obtain pure V_2O_3 film, we used only VCl_3 as the precursors with the same growth condition. To obtain V_2O_3 film (no W), we used only VCl_3 and sulfur as the precursors with the same growth condition.”*

It is a bit difficult to understand the difference between “pure V_2O_3 film” and “ V_2O_3 film (no W)” without referring to this sentence, and it may cause misunderstanding when reading the article.

Response 9. We appreciate the reviewer very much for these very helpful suggestions. Following your suggestions, we have renamed the V_2O_3 film as follows:

“To obtain the V_2O_3 -No Nuclei film, we used only VCl_3 as the as the precursors with the same growth condition. To obtain V_2O_3 -Nuclei film, we used only VCl_3 and sulfur as the precursors with the same growth condition.”

Accordingly, we have made corresponding changes and highlighted these changes in the revised manuscript (Page 9-11, Page 15-16) and supplementary information (Supplementary Figs. 16-19).

Comment 10. *The authors tried to propose a growth mechanism in which the formation of V_2O_3 film and VS_2 domains are important for the 1T WS_2 growth.*

However, it is difficult to tell if the V_2O_3 is formed during the synthesis or during the characterization, as the V compounds (VS_2 , VCl_3) are extremely air sensitive. If the characterization was done in air, oxidation can not be avoided, thus it may lead to misunderstanding in the mechanism study.

Response 10. We appreciate the reviewer very much for these valuable comments. We agree with the reviewer that V compounds such as VS_2 and VCl_3 are extremely air sensitive, especially for VCl_3 . Considering the sensitivity of VCl_3 to the air and moisture, the VCl_3 was stored in the glovebox before it was used. In this work, many characterizations for V_2O_3 film have been done to confirm the crystal structure, for example, STEM image in Fig. 2h, XRD patterns in Supplementary Fig. 2 and Raman spectra in Supplementary Fig. 10. The STEM image of 1T@ V_2O_3 /sapphire sample in Fig. 2h was acquired in the vacuum condition without heating process. The crystal structure of V_2O_3 film was clearly identified on the surface of sapphire substrate, confirming that the V_2O_3 structure was not caused by the oxidation of VS_2 or VCl_3 . The XRD patterns (Fig. S2) and Raman spectra (Fig. S10) were conducted at room temperature without heating process. From the prominent peak of V_2O_3 (006) in Fig. S2, the formed V_2O_3 film was single crystalline structure, which could not be transformed by the VS_2 or VCl_3 at the room temperature. Importantly, it was well-known that Raman spectroscopy was a very sensitive technique to detect the surface structure of the nanomaterials. If the V_2O_3 film was gradually transformed by the VS_2 or VCl_3 at room temperature, typical signals of VS_2 or VCl_3 would emerge. However, no signals that assigned to VS_2 or VCl_3 were observed, demonstrating that the Raman signals of V_2O_3 were not caused by the oxidation of VS_2 or VCl_3 . Through a series of careful analyses, we could confirm that the V_2O_3 film was formed during the synthesis and the proposed growth mechanism of 1T- WS_2 was reliable based on experimental and simulated results.

Comment 11. *The authors claim that it is a controllable approach to synthesize 1T WS_2 . Regarding the controllability, could the author control the 2H/1T ratio through the synthesis, not post annealing?*

Response 11. We appreciate the reviewer very much for the valuable question. To demonstrate the thermal stability of 1T-WS₂ monolayers, post annealing with different temperatures were conducted, as shown in Fig. 3e and Supplementary Fig. 14. Moreover, it was found that the amount of VCl₃ played an important role in controlling the 2H/1T ratio in our exploration of controllable 1T-WS₂ growth, as shown in Supplementary Fig. 23-26. It was observed that a high 2H/1T ratio was achieved when the amount of VCl₃ was ~0.02 g and only 13% 1T-WS₂ was obtained according to the XPS analysis (Supplementary Fig. 26). When the amount of VCl₃ was in the range of 0.05 g~0.08 g, dominated 2H-WS₂ monolayers were grown, as demonstrated in Supplementary Fig. 24 (olive and purple plots) and Supplementary Fig. 26. A low 2H/1T ratio was obtained when 0.03 g VCl₃ was used as the co-precursors according to the XPS analysis in Fig. 3c (91% 1T phase).

The above-mentioned Supplementary Fig. 23-26 has been added in the revised supporting information as follows:

Supplementary Fig. 23 Optical micrographs of as-prepared products using different amount of VCl₃. a, 0.01g; b, 0.02 g; c, 0.03 g; d, 0.05 g; e, 0.08 g; f, 0.10 g. T₁=860 °C, T₂ = 160 °C, Ar/H₂ = 80 sccm/20 sccm, t = 15 min; T₁ refers to the heating temperature of the furnace, T₂ refers to the heating temperature of the sulfur powder, and t refers to the growth time. All the other experimental parameters are the same. Scale bars: 5 μm.

Supplementary Fig. 24 The Raman spectra of as-prepared products using different amount of VCl_3 .

Supplementary Fig. 25 The Raman spectra of as-prepared V_2O_3 film using different amount of VCl_3 . The Raman spectra were measured on the areas where no WS_2 domains appeared.

Supplementary Fig. 26 High resolution XPS spectra of W 4f (a) and S 2s (b) core level peak regions for WS₂ domains using different amount of VCl₃. The fitting blue and pink curves represent the contributions of 1T and 2H phases, respectively. Please note that obtained WS₂ domains were transferred on the sapphire substrates for the XPS measurements.

Accordingly, the growth details and related discussion were added in the revised supplementary information as follows:

“In order to demonstrate the amount of VCl₃ has a great impact on the phase purity of WS₂, we have prepared WS₂ samples using different mass amount of VCl₃ (from 0.01 g to 0.10 g).... Whereas, the density of VS₂ nuclei was the constraining factor to impact the 1T-WS₂ formation when the amount of VCl₃ was less than 0.03 g.”

Comment 12. *In addition, this approach replies on the introduction of V and the formation of 1T-VS₂ and V₂O₃ based on the proposed mechanism, could the other V precursors, such as vanadocene, give the same effect? What will happen if you change the VCl₃ amount? Do you have the tunability towards the V concentration without changing the 1T character of WS₂?*

Response 12. We appreciate the reviewer very much for these valuable questions. According to your questions, we have made corresponding experiments to deeply understand the 1T-WS₂ growth. **Firstly**, we have tried to use vanadocene as the co-precursors and found that V SACs@1T-WS₂ monolayers were also achieved using a modified condition. It was well-known that vanadocene was commonly used in the organometallic chemical vapor deposition (OMCVD) due to its less stability and

non-toxicity. However, vanadocene is more moisture- and oxygen-sensitive compared to VCl_3 . To obtain the 1T- WS_2 domains, different amounts of vanadocene have been investigated. It was found the optimized amount of vanadocene was ~ 0.08 g. The characterizations of V SACs@1T- WS_2 monolayers were shown in Supplementary Fig. 30. From the optical micrograph, the obtained 1T- WS_2 monolayers showed similar morphology with Fig. 2b. The atomic structure of V SACs@1T- WS_2 was displayed in Supplementary Fig. 30b, confirming the 1T phase of WS_2 . The typical metallic peaks were prominently seen in the Raman spectrum (Supplementary Fig. 30c), indicative of the metallic feature of WS_2 domains. Moreover, the negligible PL intensity of obtained WS_2 domains was also provided to demonstrate the metallic nature (Supplementary Fig. 30d). It was reported that vanadocene could be completely decomposed at less than 200°C under a H_2 atmosphere (*J. Anal. Appl. Pyrolysis*, **1996**, 36, 121), so it was easier to transport in the gaseous state than VCl_3 during the heating temperature. As a result, it required higher amount of vanadocene to obtain the 1T- WS_2 domains.

The above-mentioned Supplementary Fig. 30 has been added in the revised supporting information as follows:

Supplementary Fig. 30 **a**, Optical micrograph of 1T-WS₂ domains obtained using vanadocene (0.08 g) as the co-precursors. T₁=860 °C, T₂ = 160 °C, Ar/H₂ = 80 sccm/20 sccm, t = 15 min; T₁ refers to the heating temperature of the furnace, T₂ refers to the heating temperature of the sulfur powder, and t refers to the growth time; **b**, STEM image of obtained V SACs@1T-WS₂ monolayer. The single V atoms are marked by the white dashed circles; **c**, Raman spectra of 1T/V₂O₃ (black plot) and V₂O₃ film (red plot), respectively; **d**, PL spectrum of 1T/V₂O₃ film. Scale bars: a, 20 μm; b, 1 nm.

Accordingly, we have added the description of using vanadocene as co-precursors for the 1T-WS₂ growth on Page 11 in the revised manuscript as follows:
 “Moreover, vanadocene precursors were also investigated to enrich the growth method of 1T-WS₂ monolayers (Figure S30).”

Additionally, the growth details were added in Supplementary Fig. 30 in the revised supplementary information as follows:

“Apparently, 1T-WS₂ domains were also achieved if using vanadocene as the co-precursors. ...As a result, it required higher amount of vanadocene to obtain the

1T-WS₂ domains.”

Secondly, we have prepared the WS₂ domains using different amount of VCl₃, as shown in Supplementary Fig. 23-26. As replied to the comment 11, it was found that the amount of VCl₃ could significantly affect the controllable phase growth of WS₂. The details were seen in the Supplementary Fig. 23-26 and corresponding discussion part.

Finally, the tunability towards the V concentration without changing the 1T character of WS₂ was performed by varying the heating temperatures (from 840~865 °C). Generally, two aspects could be considered to tune the V concentration during the heating process. One way is changing the amount of VCl₃, which could control the density of VS₂ nuclei formation. However, as demonstrated in Supplementary Fig. 22-26, either 2H phase or 1T phase was obtained by varying the amount of VCl₃. The other way is to tune the heating temperatures (from 840~865 °C), as the density of VS₂ nuclei could also be slightly affected by the heating temperature. As expected, the WS₂ samples with different lateral sizes were formed under the different heating temperatures in Supplementary Fig. 27. Impressively, the typical metallic peaks (J₁, J₂, J₃) were all observed in the Raman spectra of Supplementary Fig. 28, indicative of the formation of 1T-WS₂. Moreover, the atomic structures of corresponding V SACs@1T-WS₂ samples were shown in Supplementary Fig. 29, which all manifested the 1T structures of WS₂. In particular, the single V atoms were 8.0 at% (4.0 wt%), 5.0 at% (2.5 wt%), 4.0 at% (2.0 wt%) and 4.2 at% (2.1 wt%), respectively, when changing the heating temperatures from 840 °C to 865 °C, revealing that the V concentration in the 1T-WS₂ monolayers could be tuned by the heating temperature.

The above-mentioned Supplementary Fig. 27-29 have been added in the revised supporting information as follows:

Supplementary Fig. 27 1T-WS₂ domains prepared at different heating temperatures (T_1). **a**, $T_1=840$ °C; **b**, $T_1=850$ °C; **c**, $T_1=860$ °C; **d**, $T_1=865$ °C. T_1 refers to the heating temperature of the furnace, T_2 refers to the heating temperature of the sulfur powder, and t refers to the growth time. All the other experimental parameters are the same except T_1 . $T_2 = 160$ °C, Ar/H₂ = 80 sccm/20sccm, $t = 15$ min. Scale bars: 10 μm .

Supplementary Fig. 28 The Raman spectra of WS₂ domain prepared at different heating temperatures of furnace.

Supplementary Fig. 30 STEM images of V SACs@1T-WS₂ monolayers prepared at different heating temperatures. Single V atoms are marked by the white dashed circles. Scale bars: 1 nm.

Accordingly, we have added the description of influence of heating temperatures in the 1T-WS₂ growth on page 11 in the revised manuscript as follows:

“The influence of heating temperature on the synthesis of 1T-WS₂ has also been investigated, as displayed in Figure S27-29.”

Additionally, the growth details were added in the revised supplementary information as follows:

“As expected, the WS₂ samples with different lateral sizes were formed under the different heating temperatures in Supplementary Fig. 27. ...In particular, the single V atoms were 8.0 at% (4.0 wt%), 5.0 at% (2.5 wt%), 4.0 at% (2.0 wt%) and 4.2 at% (2.1 wt%), respectively, when changing the heating temperatures from 840 °C to 865 °C, revealing that the V concentration in the 1T-WS₂ monolayers could be tuned by the heating temperature.”

Comment 13. *For the HER measurements, since the VWS_2 is not a continuous film, how did the authors calculate the mass loading and the area? It seems that the author assume the film is a continuous film, which abbeys the characterization shown in Figure 2.*

Response 13. We appreciate the reviewer very much for these questions. **Firstly**, we assumed the WS_2 electrocatalyst was a continuous film in the original manuscript, because the density of as-grown 1T- WS_2 monolayers was different in the different growth locations. As shown in Supplementary Fig. 30, the stitching optical micrograph with large area was provided, indicating the WS_2 continuous film and isolated monolayers. As a result, it was difficult to accurately calculate the actual area of WS_2 monolayers on the substrate. Moreover, it was also assumed that the exfoliated WS_2 sub-monolayers were continuous film (*Nat. Mater.* **12**, 850-855 (2013)). Similar with our case, in the original manuscript, we assumed the as-grown continuous monolayer WS_2 film was deposited onto a substrate with surface area of $A_{sub.}$, then the area of A_{WS_2} was equal to the geometric area of $A_{sub.} \cdot m_{WS_2} / \rho_{WS_2} \cdot V_{WS_2} / H_{WS_2}$ represent the mass amount of WS_2 , density of WS_2 , volume of WS_2 , and monolayer thickness of WS_2 , respectively. The mass loading of continues WS_2 film was calculated according to the equation as follows:

$$\text{Mass loading of } WS_2 = \frac{m_{WS_2}}{A_{sub.}} = \frac{\rho_{WS_2} V_{WS_2}}{A_{sub.}} = \frac{\rho_{WS_2} A_{WS_2} H_{WS_2}}{A_{sub.}} = \rho_{WS_2} H_{WS_2}$$

Following your comment, we strongly agree with your viewpoint that assuming of continuous WS_2 film in this work was not accurate. Apparently, the assuming obeyed the characterization in Fig. 2. In order to evaluate the actual area of WS_2 monolayers more accurately, we made statistics for the actual area of 1T- WS_2 monolayers in Figure. S31. The continuous film was formed on the relatively down-stream of isolated WS_2 monolayers. Six representative areas were selected for the statistical analyses, as shown in Supplementary Fig. 32. The largest $A_{WS_2}/A_{sub.}$ ratio in these Figs was ~ 1.0 from Fig. S32a due to the approximately continuous film. As for the Fig. S32b-32c, the ratio of $A_{WS_2}/A_{sub.}$ was apparently less than 1.0. In particular, the relatively lower ratios of $A_{WS_2}/A_{sub.}$ were shown in Fig. S32d-S32f

according to the lower distribution density of WS₂ monolayers. To obtain the accurate $A_{WS_2}/A_{sub.}$ ratios of Fig. S32d-32e, the diameter distribution range of WS₂ monolayers was obtained by average diameter analysis software. The corresponding distribution sizes of WS₂ monolayers were displayed in Fig. S33a-S33c. According to the calculated area results in Table S4-S6. The $A_{WS_2}/A_{sub.}$ ratios were 0.26, 0.27 and 0.28, which were calculated, respectively, as follows:

$$\begin{aligned}
 A_{WS_2}/A_{sub.} \text{ ratio in Fig. S32d} &= \frac{\text{area of } WS_2 \text{ monolayers in Table S4}}{\text{area of Fig. S32d}} \\
 &= \frac{36866.6 \mu m^2}{374 \times 374 \mu m^2} = 0.26 \\
 A_{WS_2}/A_{sub.} \text{ ratio in Fig. S32e} &= \frac{\text{area of } WS_2 \text{ monolayers in Table S5}}{\text{area of Fig. S32e}} \\
 &= \frac{37908.5 \mu m^2}{374 \times 374 \mu m^2} = 0.27 \\
 A_{WS_2}/A_{sub.} \text{ ratio in Fig. S32f} &= \frac{\text{area of } WS_2 \text{ monolayers in Table S6}}{\text{area of Fig. S32f}} \\
 &= \frac{38943.7 \mu m^2}{374 \times 374 \mu m^2} = 0.28
 \end{aligned}$$

Consequently, we take the mean value of 0.27 as the lowest $A_{WS_2}/A_{sub.}$ ratio. The actual mass loading of WS₂ monolayers was in the range of 1.8-6.5 $\mu g/cm^2$, which was calculated as follows:

$$\begin{aligned}
 \text{Lowest Mass loading of } WS_2 &= \frac{m_{WS_2}}{A_{sub.}} = \frac{\rho_{WS_2} V_{WS_2}}{A_{sub.}} = \frac{\rho_{WS_2} A_{WS_2} H_{WS_2}}{A_{sub.}} = \\
 &0.27 \rho_{WS_2} H_{WS_2} = 1.8 \mu g/cm^2 \\
 \text{Largest Mass loading of } WS_2 &= \frac{m_{WS_2}}{A_{sub.}} = \frac{\rho_{WS_2} V_{WS_2}}{A_{sub.}} = \frac{\rho_{WS_2} A_{WS_2} H_{WS_2}}{A_{sub.}} = \\
 &\rho_{WS_2} H_{WS_2} = 6.5 \mu g/cm^2
 \end{aligned}$$

Rigorously, we have revised the mass loading value in the revised manuscript, which has been highlighted on Page 12 as follows:

“...1.8~6.5 $\mu g/cm^2$ (see details in Fig. S31-S33 and Table S4-S6),”

We have deleted the description of “approximately one continuous layer of WS₂ over the electrode surface area to 6.5 $\mu g/cm^2$ ” in the “Electrochemical Measurements” section to avoid any confusion, which has been highlighted on Page 18 in the revised manuscript.

We have deleted the annotation of “6.5 $\mu\text{g cm}^2$ ” in the Supplementary Fig. S34 to avoid any confusion in the revised supplementary information.

Additionally, we have added one sentence of “TOF values were calculated using the mass loading of 6.5 $\mu\text{g/cm}^2$.” in the Table S7 to avoid any confusion, which has been highlighted in Table S7 in the revised supplementary information. **Please note that the original TOF values in Table S7 were all underestimated according to the TOF calculation method on Page 19 in the revised manuscript. Despite using the overestimated area of WS_2 , the obtained TOF values of V SACs@1T- WS_2 at different overpotentials still surpassed that most of the reported single metals catalysts.**

The above-mentioned Supplementary Fig. 31-33 and Table S4-S6 have been added in the revised supporting information as follows:

Supplementary Fig. 31 Stitching optical micrograph of as-grown V SACs@1T- WS_2 .
Scale bar: 500 μm .

Supplementary Fig. 32 Six representative optical micrographs taken from Fig. S31. Scale bars: 50 μm .

Supplementary Fig. 33 a-c, Statistics of WS_2 distribution with different lateral sizes in Fig. S32d (a), Fig. S32e (b) and Fig. S32f (c), respectively. **d**, the calculated $A_{WS_2}/A_{sub.}$ ratio in the respective Figs.

Table S4. Statistics of WS₂ distribution with different lateral sizes in Fig. S32d and the corresponding areas.

Diameter Distribution (μm)	Mean diameter (μm)	Amount	Freq.	Area (μm ²)
11-13.4	12.2	1	1.1%	116.8
13.4-15.8	14.6	8	8.7%	1338.6
15.8-18.2	17	10	10.9%	2268.7
18.2-20.6	19.4	16	17.4%	4727.1
20.6-23	21.8	22	23.9%	8207.4
23-25.4	24.2	15	16.3%	6895.9
25.4-27.8	26.6	9	9.8%	4998.9
27.8-30.2	29	5	5.4%	3300.9
30.2-32.6	31.4	3	3.3%	2321.9
32.6-35	33.8	3	3.3%	2690.4
Total Area of WS ₂ monolayers in Fig. S32d				36866.6

*To calculate the area of WS₂ monolayers, we assume each WS₂ monolayer was circular and the mean diameter value represents the diameter (d) of circular WS₂. The area of each WS₂ monolayer was calculated according to the equation as follows:

$$\text{Area of } WS_2 \text{ monolayer} = \pi r^2 = \frac{1}{4} \pi d^2$$

Table S5. Statistics of WS₂ distribution with different lateral sizes in Fig. S32e and the corresponding areas.

Diameter Distribution (μm)	Mean diameter (μm)	Amount	Freq.	Area (μm ²)
10-13.1	11.55	1	1.3%	104.7
13.1-16.2	14.65	4	5.3%	673.9
16.2-19.3	17.75	8	10.5%	1978.6
19.3-22.4	20.85	15	19.7%	5118.9
22.4-25.5	23.95	15	19.7%	6754.2
25.5-28.6	27.05	17	22.4%	9764.6
28.6-31.7	30.15	8	10.5%	5708.7
31.7-34.8	33.25	4	5.3%	3471.5
34.8-37.9	36.35	3	3.9%	3111.7
37.9-41	39.45	1	1.3%	1221.7
Total Area of WS ₂ monolayers in Fig. S32e				37908.5

Table S6. Statistics of WS₂ distribution with different lateral sizes in Fig. S32f and the corresponding areas

Diameter Distribution (μm)	Mean diameter (μm)	Amount	Freq.	Area (μm ²)
13-15.3	14.15	1	1.2%	157.2
15.3-17.6	16.45	6	7.4%	1274.5
17.6-19.9	18.75	6	7.4%	1655.9
19.9-22.2	21.05	14	17.3%	4869.7
22.2-24.5	23.35	11	13.6%	4708.0
24.5-26.8	25.65	18	22.2%	9296.4
26.8-29.1	27.95	17	21.0%	10425.1
29.1-31.4	30.25	3	3.7%	2155.0
31.4-33.7	32.55	3	3.7%	2495.1
33.7-36	34.85	2	2.5%	1906.8
Total Area of WS ₂ monolayers in Fig. S32f				38943.7

REVIEWER COMMENTS

Reviewer #2 (Remarks to the Author):

The authors have well addressed my concerned issues.

Reviewer #3 (Remarks to the Author):

The authors addressed part of the comments. However, the main issue now is the detection of V by XPS, and the possible oxidation of the samples due to the presence of V. If the V content ranges from 4at.% to 8at.% in WS₂ monolayer, XPS should not have any problem in detecting V. In addition, a careful study of O and S by XPS and EELS is needed to truly confirm that substitutional V is bonded to S, instead of forming VO₂/V₂O₃. Therefore, I would encourage the authors to address these points before I can recommend publication of this work. Detailed comments are below:

1. In Figure 2c, could the authors provide the intensity profile of V SACs@1T-WS₂ with V atoms, and compare it with the simulation? Figure 2f only shows the line profile with W and S atoms. The profiles with V, W, and S can help distinguish the V@W sites to vacancy@W sites.
2. The authors stated that a modified PMMA-assisted transfer process was applied to prepare V SACs@1T-WS₂ samples for XPS, STEM and EELS characterizations. However, it is my understanding that this transfer method is based on mild chemical etching followed by mechanical delamination. Thus, I am still not convinced that the transfer process can detach V SACs@1T-WS₂ from V₂O₃ and avoid the presence of vanadium oxides on various characterizations and HER tests. In addition, although V³⁺ state was not observed in EELS, it is noted that EELS can only reveal local coordination states at the nanometer scale, and may not be representative to claim the absence of V₂O₃ in the entire sample. Even if V₂O₃ was removed during the transfer process, other compounds such as VO₂ or VS₂ could be present. How did the authors rule out the possibility of VO₂/VS₂ within the samples? Could the authors provide the EELS and XPS of O and S as well?
3. The authors claimed that no V signal can be detected by XPS. However, on page 6, the authors mentioned that V is 4at%. XPS's detection limit should be lower than that, and the technique is able to detect 4at% V in WS₂. Similar work can be found in "Monolayer Vanadium-doped Tungsten Disulfide: A Room-Temperature Dilute Magnetic Semiconductor". In that work, 1.5at% substitutional V can be detected in WS₂ monolayer. Could the authors provide the XPS of survey and O as well?
4. If for any reason, the authors still cannot detect V by XPS on the transferred sample, could the authors deconvolute the S 2p peak to see if there is any V-S bond formation?
5. The authors claimed the stability under ambient conditions, which is confirmed by the Raman. J1, J2, J3 peaks found samples stored under ambient conditions for a year. However, one main issue for V-doped samples is the stability against oxidation. Thus, XPS is recommended to identify V, O, S, and W for the samples.
6. Since HER was performed after transferring the film to glassy carbon substrate. How did the authors ensure a good adhesion between the film and the substrate? and avoid flakes from peeling off from the glassy carbon? For stability tests, amperometric I-t stability curves is recommended.
7. For TOF calculation, the authors only considered V as the active sites, thus it may overestimate the TOF. However, W edges, vacancies, defects in WS₂ are also considered as the active sites. Thus, the TOF calculation seems not that accurate. ECSA is recommended to quantify the electrochemical active area and quantify the active sites. It would be better to compare the ECSA between samples with/without V as well.

8. Based on Fig. S31, both film and flakes can be found in one sample. Does that mean the sample morphology is not uniform? As can be seen in Fig.2a, it is clear that the color is not uniform across the substrates. Could the authors comment on the homogeneity of the sample? In addition, for film region, is it still monolayer with similar V concentration?

Reviewer #3 (Remarks to the Author):

Comment. *The authors addressed part of the comments. However, the main issue now is the detection of V by XPS, and the possible oxidation of the samples due to the presence of V. If the V content ranges from 4at.% to 8at% in WS₂ monolayer, XPS should not have any problem in detecting V. In addition, a careful study of O and S by XPS and EELS is needed to truly confirm that substitutional V is bonded to S, instead of forming VO₂/V₂O₃. Therefore, I would encourage the authors to address these points before I can recommend publication of this work. Detailed comments are below:*

Response. We appreciate the reviewer very much for his/her comments. His/Her concerns are addressed as follows.

Comment 1. *In Figure 2c, could the authors provide the intensity profile of V SACs@1T-WS₂ with V atoms, and compare it with the simulation? Figure 2f only shows the line profile with W and S atoms. The profiles with V, W, and S can help distinguish the V@W sites to vacancy@W sites.*

Response 1. We appreciate the reviewer very much for the very good suggestion. According to your suggestion, we have made simulated STEM images with both V SACs@1T-WS₂ and W_{vac}@1T-WS₂ to address this comment. The simulated STEM images were obtained with the help of experts from Prof. Gu Lin's group in China. As shown in Fig. S6-S7, the V elemental identity in 1T-WS₂ from different line profile sequences were clearly verified as compared to the simulated W_{vac}@1T-WS₂.

Accordingly, the V elemental identities from different line profile sequences have been added in the revised manuscript to address this comment, which has been highlighted on Page 6 as follows:

“Additionally, W-S-S-V-S-S-W and W-V-W intensity profile sequences from both experimental and simulated STEM images are also achieved to verify the V atoms replacement at W sites (see details in Fig. S6-S7).”

The above-mentioned Supplementary Fig. 6 and Supplementary Fig. 7 have been added in the revised supporting information as follows:

Supplementary Fig. 6 a-d, Experimental (a) and simulated (c) STEM images of V SACs@1TWS₂, respectively. Corresponding experimental (b) and simulated (d) intensity sequence profiles of W-S-S-V-S-S-W (red dashed arrow) indicated by the STEM images. The white dashed circles represent the V atoms; **e**, Simulated STEM image of 1T-WS₂ with vacancies at W sites (W_{vac}@1T-WS₂). The purple dashed circles represent the vacancies at W sites; **f**, Corresponding intensity sequence profile of W-S-S-W_{vac}-S-S-W (red dashed arrow indicated by Figure S6e).

Supplementary Fig. 7 a-d, Experimental (a) and simulated (c) STEM images of V SACs@1TWS₂, respectively. Corresponding experimental (b) and simulated (d) intensity sequence profiles of W-V-W (yellow dashed arrow) indicated by the STEM images. The white dashed circles represent the V atoms; e, Simulated STEM image of W_{vac}@1T-WS₂. The purple dashed circles represent the vacancies at W sites; f, Corresponding intensity sequence profile of W-W_{vac}-W (yellow dashed arrow indicated by Figure S7e).

Comment 2. *The authors stated that a modified PMMA-assisted transfer process was applied to prepare V SACs@1T-WS₂ samples for XPS, STEM and EELS characterizations. However, it is my understanding that this transfer method is based on mild chemical etching followed by mechanical delamination. Thus, I am still not convinced that the transfer process can detach V SACs@1T-WS₂ from V₂O₃ and avoid the presence of vanadium oxides on various characterizations and HER tests. In addition, although V³⁺ state was not observed in EELS, it is noted that*

EELS can only reveal local coordination states at the nanometer scale, and may not be representative to claim the absence of V_2O_3 in the entire sample. Even if V_2O_3 was removed during the transfer process, other compounds such as VO_2 or VS_2 could be present. How did the authors rule out the possibility of VO_2/VS_2 within the samples? Could the authors provide the EELS and XPS of O and S as well?

Response 2. We appreciate the reviewer very much for these insightful questions. **Firstly**, in this work, diluted HF (5%) was used to transfer the V SACs@1T- WS_2 from the surface of V_2O_3 /sapphire. And similar transferring methods have been widely used in many previously published literatures^{1,2}. Specially, the V_2O_3 film formed in our CVD growth is very high-quality with single crystalline structure, which has been demonstrated by the cross-sectional STEM image in Fig. 2g-2h and XRD patterns in Fig. S2. As the crystal structure of V_2O_3 was similar with Al_2O_3 , the surface of V_2O_3 film was very tough, which was not easy to be detached with V SACs@1T- WS_2 /PMMA film during the transfer process. As a result, the V_2O_3 film was still on the sapphire substrate after transferring, but not removed during the transferring process.

Secondly, we totally agree with the reviewer that EELS can only reveal local coordination states at the nanometer scale and the results of V signals detected by EELS may not be representative to claim the absence of V_2O_3 , VS_2 and VO_2 in the entire sample. To further confirm the absence of $V_2O_3/VO_2/VS_2$ in the entire transferred 1T sample, XPS spectra of transferred 1T sample on the highly oriented pyrolytic graphite (HOPG) substrate were obtained. As the oxygen in the Al_2O_3 or SiO_2/Si wafer would disturb the detection of O in the V_2O_3/VO_2 . Hence, the choice of HOPG substrate was very necessary. Initially, we have measured the Raman spectrum to demonstrate the successful transferring of V SACs@1T- WS_2 sample on the HOPG substrate, which was shown in Figure R1. The XPS spectra of V SACs@1T- WS_2 sample on the HOPG substrate was shown in Figure S10-S11. Except for the signals of W4f and S 2p, the V 2p signals were detected after a long-term acquisition time during the XPS scanning cycles with much weaker intensity than W 4f (Fig. S11a)

and S 2p (Fig. S11b), which should be caused by the low atomic density of V atoms in the 1T sample. After deconvoluting the V 2p signals, the peaks at 516.4 eV and 523.5 eV were assigned to V^{4+} from V-S bond in 1T sample.³⁻⁷ Please note that if VS_2 was present in the transferred sample, a prominent V 2p peak should be detected. However, only weak V signals were detected, indicating no VS_2 contamination in the transferred sample.

Importantly, if the V-based contaminations (e.g. V_2O_3 , VO_2 , VS_2) were contained in the transferred 1T sample, obviously enhanced V/W molar ratio would be measured. However, the molar ratio of W/V was $\sim 95:5$ from the XPS analyses, which was close to the result obtained by the STEM image (~ 4 at%), revealing the absence of V-based contaminations in the transferred 1T sample. In addition, only C-O bond at 532.4 eV was obtained and no V-O bond from vanadium oxidation (530.0 eV)⁸ was observed from the O 1s peak, further confirming no V_2O_3 or VO_2 in the transferred 1T sample. The C-O bond should be caused by the slightly oxidation of HOPG on the surface during the transferring process.

Finally, the EELS and XPS of O and S were displayed in Fig. S5 and Fig. S18, respectively. The EELS result of S was in consistent with the previously reported literature,⁹ revealing the presence of S in the transferred 1T sample. No O signals were detected from the EELS spectrum in Fig. S5b, excluding the V-based oxidation contaminations in the measured sample.

According to the reviewer's suggestion, the high-resolution of O 1s in the transferred 1T sample on the sapphire substrate was also shown in Fig. S18b. However, the signal of O 1s was ascribed to the sapphire substrate. Additionally, the high-resolution of S 2p was already provided in Fig. 3c (right) in the manuscript.

Accordingly, the absence of V-based contaminations (VS_2 , V_2O_3 , VO_2) has been demonstrated by XPS spectra in the revised manuscript to address this comment, which has been highlighted on Page 6 as follows:

“To exclude the presence of V-based contaminations (eg. V_2O_3 , VO_2 and VS_2) in the transferred 1T sample, XPS spectra of 1T sample transferred on highly oriented pyrolytic graphite (HOPG) were performed (see details in Figure S10-S11).”

Accordingly, the discussion of absence of V-based contaminations (VS_2 , V_2O_3 , VO_2) has been added in the revised supporting manuscript to address this comment, which has been highlighted in Supplementary Fig. 11 as follows:

“To exclude the possibility of VS_2 and V_2O_3/VO_2 contaminations in the transferred V SACs@1T- WS_2 monolayers, we transferred V SACs@1T- WS_2 monolayers on the HOPG substrate for the further XPS analyses....

...In addition, only C-O bond at 532.4 eV was obtained and no V-O bond from vanadium oxidation (530.0 eV)⁷ was observed from the O 1s peak, further excluding the possibility of V_2O_3 or VO_2 in the transferred 1T sample. The C-O bond should be caused by the slightly oxidation of HOPG on the surface during the transferring process.”

According to your suggestion, both the survey scan and high resolution XPS spectra are provided, as shown in following Supplementary Fig. 10 and Supplementary Fig. 11 in the revised supporting information.

Figure R1 a, Optical micrograph of V SACs@1T- WS_2 domains transferred on HOPG substrate. The white arrows indicate the 1T samples while the yellow arrows represent the contaminations caused by the PMMA residue in transferring process; b, Raman spectra of the V SACs@1T- WS_2 sample. Scale bars: a, 10 μm .

Supplementary Fig. 10 Survey scan of V SACs@1T-WS₂ transferred on HOPG substrate.

Supplementary Fig. 11 a-d, High-resolution XPS spectra of W 4f (a), S 2p (b), V 2p (c) and O 1s (d) core levels of V SACs@1T-WS₂ transferred on HOPG substrate. The fitting blue and pink curves represent the contributions of 1T and 2H phases in Fig.

S11 a-b, respectively. The molar ratio of W/V obtained by the XPS analysis was ~95:5.

Accordingly, the description of EELS of S and O has been added in the revised manuscript to address this comment, which has been highlighted on Page 6 as follows:

“The EELS of S and O spectra in Figure S5 were shown to further reveal the presence of S and absence of O in the transferred 1T samples, excluding the V signals from V-based oxidations.”

According to your suggestion, EELS of O and S are provided in the revised supplementary manuscript, as shown in following Supplementary Fig. 5 in the revised supporting information.

Supplementary Fig. 5 EELS spectrum of S (a) and O (b) in the V SACs@1T-WS₂ monolayer. The carbon detection was associated with remaining PMMA polymer during the transferring process.

Accordingly, the description of survey scan of V SACs@1T-WS₂ and high-resolution of O 1s on the fresh sapphire substrate has been added in the revised manuscript to address this comment, which has been highlighted on Page 9 as follows:

“The survey scan of V@ SACs@1T-WS₂ was shown in Figure S18a. The O 1s signal in Figure S18b was ascribed to the sapphire substrate.”

According to your suggestion, the survey scan of V SACs@1T-WS₂ and

high-resolution of O 1s on the fresh sapphire substrate are provided in the revised supplementary manuscript, as shown in following Supplementary Fig. S18 in the revised supporting information.

Figure S18. **a**, Survey scan of the V SACs@1T-WS₂ sample transferred on the sapphire substrate; **b**, High-resolution of XPS spectrum of O 1s.

Comment 3. *The authors claimed that no V signal can be detected by XPS. However, on page 6, the authors mentioned that V is 4at%. XPS's detection limit should be lower than that, and the technique is able to detect 4at% V in WS₂. Similar work can be found in “Monolayer Vanadium-doped Tungsten Disulfide: A Room-Temperature Dilute Magnetic Semiconductor”. In that work, 1.5at% substitutional V can be detected in WS₂ monolayer. Could the authors provide the XPS of survey and O as well?*

Response 3. We appreciate the reviewer very much for these insightful questions. We also appreciate the reviewer for bringing us this very important paper about V-doped 2H-WS₂ monolayers to our attention. We have cited this paper in the manuscript (please see reference 35). We read this related paper carefully and found that: In that paper, V-doped 2H-WS₂ on the SiO₂/Si wafer was reported. However, the growth method and application in that paper are totally different from our materials in this work (i.e., V SACs 1T-WS₂/V₂O₃ film grown on sapphire substrates). However, we are inspired by the deeply XPS analyses in that work, we found that the probable

reason that no V signals were detected in our first detection of high-resolution V 2p in the transferred sample was that the short time of acquisition time during the measurement. Hence, to figure out whether the V signals were detected, we have tried to measure the high-resolution of V 2p with long-term acquisition time. Simultaneously, it was mentioned that the reviewer has raised the questions of VS₂ and V₂O₃/VO₂ contaminations in the transferred V SACs@1T-WS₂ monolayers. As the O 1s signals were also from the sapphire substrate (α -Al₂O₃), we transferred V SACs@1T-WS₂ monolayers on the HOPG substrate for the further XPS analyses. The high-resolution XPS spectra of W 4f, S 2p, V 2p and O 1s at different acquisition time were shown in Figure R2. Except for the signals of W 4f and S 2p, the V 2p signals were detected after a long-term acquisition time during the XPS scanning with much weaker intensity than W 4f (Fig. R2a) and S 2p (Fig. R2b), which should be caused by the low atomic density of V atoms in the 1T sample. After deconvoluting the V 2p signals (see in Fig. S11c), the peaks at 516.4 eV and 523.5 eV were assigned to V⁴⁺ from V-S bond in 1T sample.³⁻⁷ Importantly, if the V-based contaminations (e.g. V₂O₃, VO₂, VS₂) were contained in the transferred 1T sample, obviously enhanced V/W molar ratio would be measured. However, the molar ratio of W/V was ~95:5 from the XPS analyses, which was close to the result obtained by the STEM image (~4 at%), revealing the absence of V-based contaminations in the transferred 1T sample. Importantly, Only C-O bond at 532.4 eV was obtained and no V-O bond from vanadium oxidation (530.0 eV)⁸ was observed from the O 1s peak, further confirming no V₂O₃ or VO₂ were contained in the transferred 1T sample. The C-O bond should be caused by the slightly oxidation of HOPG on the surface during the transferring process.

Accordingly, the detection of V signals has been demonstrated by XPS spectra in the revised manuscript to address this comment, which has been highlighted on Page 7 as follows:

“The V signal in Fig. S11c was ascribed to the V-S bond⁴¹, which was consistent with the EELS result in Fig. 2e.”

According to your suggestion, both the survey scan and high resolution XPS

spectra are provided as Supplementary Fig. S10 and Supplementary Fig. S11 in the revised supplementary manuscript, please refer to the response to Comment 2.

Figure R2 High-resolution XPS spectra of W 4f (a), S 2p (b), V 2p (c) and O 1s (d) core levels of V SACs@1T-WS₂ transferred on HOPG substrate. The spectra were obtained at different acquisition time.

Comment 4. *If for any reason, the authors still cannot detect V by XPS on the transferred sample, could the authors deconvolute the S 2p peak to see if there is any V-S bond formation?*

Response 4. We appreciate the reviewer very much for the good suggestion. As demonstrated in Figure R2, the V signals were detected after longtime scanning cycles during the XPS analyses. And after deconvoluting the V 2p peak, the signal of V 2p at 516.4 eV was assigned to V⁴⁺, which should be from the V-S bond formation in the V SACs@1T-WS₂. Please note that the S 2p peak that assigned to the V-S bond (S 2p_{3/2}: 162.1 eV and S 2p_{1/2} 163.3 eV)⁷ was overlapped to S 2p peak in the 2H-WS₂¹⁰. And the S signals are dominantly ascribed to the W-S bond in the 1T-WS₂ sample, hence, it

is difficult to get accurate V-S bond information from the S 2p peak.

Comment 5. *The authors claimed the stability under ambient conditions, which is confirmed by the Raman. J_1 , J_2 , J_3 peaks found samples stored under ambient conditions for a year. However, one main issue for V-doped samples is the stability against oxidation. Thus, XPS is recommended to identify V, O, S, and W for the samples.*

Response 5. We appreciate the reviewer very much for the good suggestion. We agree the reviewer that the V-doped samples were not stable. That's why obviously enhanced intensity of E_{2g}^1/A_{1g} was seen in Fig. S19, indicative of a decreasing 1T phase. As it was known that the surface of V_2O_3 film could be gradually oxidated upon exposure to the ambient condition. According to your suggestion, XPS spectra of 1T sample without transferring (after one year) were obtained to identify the oxidation degree of the 1T sample and V_2O_3 film, as shown in Fig. S20. The wide-scan spectrum was shown in Fig. S20a and the main components are O, V, W, S, Al (from the sapphire). A high amount of 1T phase is still preserved even after one year (~60%) from Fig. S20b, indicating the durable stability of 1T sample. Meanwhile, the V3p was found in Fig. S20b, which could not be used to analyze the V-base oxidation states caused by a complicated interaction between V3p and 3d electrons.⁸ However, three contributions were found in the V 2p XPS spectrum, as shown in Fig. S20d: V^{3+} , V^{4+} , V^{5+} , respectively, at 514.47 eV, 516.19 eV and 517.40 eV.⁸ The main component is V^{5+} , which should be caused by the oxidation of V_2O_3 film on the surface in the air. The V^{4+} signal from V-S bond in the 1T sample was also involved due to the overlapped peak position of VS_2 and VO_2 . Please note that no VO_2 or V_2O_5 signals were detected by the Raman from Fig. S19 (red plot), which should be caused by overwhelming signals from 1T sample and V_2O_3 film on the surface of sample.

Accordingly, the XPS spectra for the 1T/ V_2O_3 on the sapphire substrate after one year have been added in the revised manuscript to address this comment, which has been highlighted on Page 9 as follows:

“Importantly, a high 1T phase (~60%) is preserved in the sample from the XPS

analysis in Figure S20b, and the decrease of 1T/2H ratio should be probably caused by the oxidation of V_2O_3 film on the surface, as demonstrated in Figure S20d.”

The above-mentioned Supplementary Fig. S20 has been added in the revised supporting information as follows:

Supplementary Fig. 20 a, Survey spectra of V SACs@1T- WS_2/V_2O_3 film grown on sapphire substrate after 1 year; **b-d**, High-resolution XPS spectra of V 3p (b), W 4f (b), S 2p (c), O 1s (d) and V 2p (d) core level peak regions. The fitting blue and pink curves represent the contributions of 1T and 2H phases in Fig. S20b-c, respectively.

The discussion of XPS spectra for the 1T/ V_2O_3 on the sapphire substrate after one year has been added in Supplementary Fig. 20 in the revised supplementary information as follows:

“The wide-scan spectrum was shown in Fig. S20a and the main components are O, V, W, S, Al (from the sapphire).Please note that no VO_2 or V_2O_5 signals were detected by the Raman from Fig. S19 (red plot), which should be caused by overwhelming signals from 1T sample and V_2O_3 film on the surface of sample.”

Comment 6. Since HER was performed after transferring the film to glassy carbon substrate. How did the authors ensure a good adhesion between the film and the

substrate? and avoid flakes from peeling off from the glassy carbon? For stability tests, amperometric I-t stability curves is recommended.

Response 6. We appreciate the reviewer very much for the good question. It was well-known that the interaction between the transferred WS₂ film and the glassy carbon electrode was weak Van der Waals's force. To enhance the adhesion and protect the film from peeling off from the electrode, appropriate Nafion film (a conductive polymer) was casted onto the surface of WS₂ film during the HER catalytic measurements. Please note that this kind of protection was widely used in many previously published literatures.¹⁰⁻¹⁵

Following your suggestion, the stability test was conducted in 0.5 M H₂SO₄ electrolyte for 100 h. As revealed by the chronoamperometric curve of V SACs 1T-WS₂ electrocatalyst in Supplementary Fig. 47, the current density for the V SACs 1T-WS₂ electrocatalyst displayed a slight current decay of 1.0 mA cm⁻² after 24 h and 3.4 mA cm⁻² after 100 h, indicating a high stability of V SACs 1T-WS₂ catalyst. Additionally, the good stability of HER performance suggested a strong adhesion between WS₂ film and glassy carbon electrode.

Accordingly, the description of the electrochemical stability test has been added in the revised manuscript to address this comment, which has been highlighted on Page 14-15 as follows:

“The stability test was conducted at current densities higher than 10 mA/cm² in 0.5 M H₂SO₄ electrolyte for 100 h. As revealed by the chronoamperometric curve of V SACs 1T-WS₂ electrocatalyst in Figure S47, the current density for the V SACs 1T-WS₂ electrocatalyst displayed a slight current decay of 1.0 mA cm⁻² after 24 h and 3.4 mA cm⁻² after 100 h, indicating a high stability of V SACs 1T-WS₂ catalyst.”

The above-mentioned Supplementary Fig. 47 has been added in the revised supporting information as follows:

Supplementary Fig. 47 Chronoamperometric curve of V SACs 1T-WS₂ catalyst at an overpotential of 400 mV in 0.5 M H₂SO₄ electrolyte.

Comment 7. For TOF calculation, the authors only considered V as the active sites, thus it may overestimate the TOF. However, W edges, vacancies, defects in WS₂ are also considered as the active sites. Thus, the TOF calculation seems not that accurate. ECSA is recommended to quantify the electrochemical active area and quantify the active sites. It would be better to compare the ECSA between samples with/without V as well.

Response 7. We appreciate the reviewer very much for these very helpful comments. **Firstly**, according to your suggestion, electrochemical surface area (ECSA) of samples with and without V were acquired by achieving the double-layer capacitance (C_{dl}). The WS₂ samples were transferred on the glassy carbon electrode for the typical cyclic voltammograms between 0.15 V 0.35 V (vs. RHE) at various scan rates (10~210 mV/s) in 0.5 M H₂SO₄, as shown in Figure S46. Half of the linear slopes obtained in Figure S46b-d were equivalent to the C_{dl} . Apparently, the C_{dl} of V SACs@1T-WS₂ was 139.5 $\mu\text{F}/\text{cm}^2$, which was much higher than that of 2H-WS₂ (61.7 $\mu\text{F}/\text{cm}^2$). The reference specific capacitance (C_s) of 40 $\mu\text{F}/\text{cm}^2$ is used in this work.¹⁶⁻¹⁹ According to the equation:

$$ECSA = \frac{C_{dl}}{C_s}$$

$$ECSA_{1T} = \frac{C_{dl\ 1T}}{C_s} = \frac{139.5 \frac{\mu F}{cm^2}}{40 \frac{\mu F}{cm^2} \times per\ cm^2} = 3.49\ cm^2$$

$$ECSA_{2H} = \frac{C_{dl\ 2H}}{C_s} = \frac{61.7 \mu F/cm^2}{40 \mu F/cm^2 \times per\ cm^2} = 1.54\ cm^2$$

Secondly, Generally, ECSA measurements only represent the electrochemically active sites on the electrodes, however, not all the electrochemically active sites are catalytically active.^{16,18-24} As a result, such kind of measurements could be better for the qualitative comparison of different electrocatalysts.^{16,18-24}

Thirdly, as pointed by the reviewer, we agree that W edges, vacancies, defects in 1T-WS₂ support are also considered as the active sites, which could overestimate the TOF values if the V atoms were only considered the active sites. The HER catalytic activity shown in Fig. 5a revealed that both V atoms and high purity 1T phase gave the contribution to the higher HER activity than 2H_{1T}-WS₂ or pure 2H-WS₂. Nevertheless, V single atoms dominantly determined the high HER performance from the DFT results, as the intrinsic activity of 1T-WS₂ was much lower than V SACs@1T-WS₂ due to the relatively larger free energy of H adsorption (ΔG_H) of 1T-WS₂ (0.28 eV) than V atom sites decorated 1T-WS₂ (0.05 eV, Table S8). Additionally, active sites from the W edge (or S edge) could be negligible due to the large free energy of H adsorption (ΔG_H) (Table S8). Importantly, the atomic density of V is much higher than the density of vacancy defects from the STEM image in both 1T sample (Fig. 2c) and 2H_{1T} (Fig. S8), indicating a dominated effect of V atoms on the enhanced HER activity of 1T-WS₂ sample.

Finally, it was widely demonstrated the defect-rich supports could stabilize higher density of SACs for better HER catalytic performance. However, to achieve a quantitative insight into the HER activity, the TOF values were generally calculated based on the single metal sites as the main active sites, even if the supports (N-doped carbon supports,^{14,25,26} 1T'-MoS₂,¹² 2H-MoS₂^{15,27} etc.) could also contribute the HER activity. Particularly, it's worth noting that the TOF values shown in this work were

taken the minimal values as we assume the V atoms are the most active sites in a continuous 1T-WS₂ monolayer support (see Table S7 and discuss section in Fig. S43).

Accordingly, the description of ECSA has been added in the revised manuscript to address this comment, which has been highlighted on Page 13 and page 20 as follows:

Page 13:

“The electrochemical surface area (ECSA) was calculated as an important factor to affect the catalytic activity of electrocatalysts.⁵³⁻⁵⁵ The calculated details were shown in the experimental section and Figure S46. C_{dl} and ECSA values were 139.5 μF/cm² and 3.49 cm² for V SACs@1T-WS₂ and 61.7 μF/cm² and 1.54 cm² for 2H-WS₂, suggesting the critical contributions of V atom sites and high purity of 1T-WS₂.”

Page 20:

“ECSA and double layer capacitance (C_{dl}) are determined by cyclic voltammograms at various scan rates (10, 30, 50, 70, 90, 110, 130, 150, 170, 190, 210 mV/s) in the potential range (0.15~0.35 V vs. RHE). The capacitive currents (ΔJ) are plotted as a function of scan rate and C_{dl} is equal to the half of slope. The reference specific capacitance (C_s) of 40 μF/cm² is used in this work. The ECSA for the different catalysts are achieved based on the following equation

$$ECSA = \frac{C_{dl}}{C_s}$$

”

The above-mentioned Supplementary Fig. 46 has been added in the revised supporting information as follows:

Supplementary Fig. 46 Voltammograms of (a) V SACs@1T-WS₂ and (c) 2H-WS₂ electrocatalysts at various scan rates (10 ~ 210 mV/s); Electrochemically active surface area (b) V SACs@1T-WS₂ and (d) 2H-WS₂ estimated from the voltammograms at various scan rates (10 ~ 210 mV/s).

Comment 8. *Based on Fig. S31, both film and flakes can be found in one sample. Does that mean the sample morphology is not uniform? As can be seen in Fig.2a, it is clear that the color is not uniform across the substrates. Could the authors comment on the homogeneity of the sample? In addition, for film region, is it still monolayer with similar V concentration?*

Response 8. We appreciate the reviewer very much for the helpful questions. **Firstly**, according to your suggestion, we have labeled the different growth regions in Fig. 2a, as shown in Fig. 38. Region A represents the V₂O₃ film without 1T sample. Region B represents the V₂O₃ film with isolated 1T flakes. Region C represents the 1T merged film caused by the higher nucleation density of WS₂ than in Region B, which was commonly reported in many literatures using CVD technique.²⁸⁻³¹ Region D represents the very downstream area, which was not able to grow V₂O₃ film and WS₂ sample on the surface.

Secondly, as shown in Fig. S39, the isolated flakes (right white arrow) in growth region B have domain sizes from 20~30 μm . On the left side in growth region C, the WS_2 domains are merged into a continuous film. This kind of non-uniformity in nucleation density and domain size is a limitation of the present CVD technique, which have been demonstrated in many previously reported literatures about TMDs grown on Si/SiO_2 ^{29,32}, quartz³³ and sapphire substrates^{28,30,31}. As a result, it is inevitable to grow isolated and merged film domains in our CVD process for 1T monolayers growth. However, each growth contained $\sim 0.5 \text{ cm} \times 2.0 \text{ cm}$ (1.0 cm^2) region where hundreds of isolated flakes grown and $\sim 0.5 \text{ cm} \times 1.0 \text{ cm}$ (0.5 cm^2) region where merged 1T film grown. To transfer the 1T samples for the investigation of HER catalytic performance, we transferred the samples at the region B (near the interface between region B and C). However, it was probable to transfer a small part of 1T merged film onto the glassy carbon electrode. To acquire a relatively accurate geometric area, we made statistics for the real geometric area of 1T samples (considering both isolated flakes and merged film, see details in Fig. S43-44).

Finally, the characterizations of 1T merged film have been provided in Fig. S40-S42. The atomic structure of V SACs@1T- WS_2 merged film were shown in Fig. S40. The images were randomly taken from four different areas in region C. The 1T phase of WS_2 was confirmed with V atoms distribution from the high-resolution STEM images. In particular, the average single V atoms was $\sim 4.1 \text{ at\%}$ (2.1 wt%), which was close to the atomic density of V atoms in the growth region B (Fig. 2c, $\sim 2.0 \text{ wt\%}$). The W-S-S-V-S-S-W intensity profile sequences were also achieved to identify the V atoms replacement at W sites, as shown in Fig. S40e. The typical metallic peaks were shown in the Raman spectrum (Supplementary Fig. 41), confirming the metallic feature of WS_2 sample. The monolayer thickness was confirmed by cross-sectional STEM images in Fig. S42.

Please note that the above-mentioned Supplementary Fig. 31 in the original supplementary information has been revised as Supplementary Fig. 39 in the revised supplementary information.

Accordingly, the above-mentioned Supplementary Fig. S38-S42 have been added

in the revised supporting information as follows:

Supplementary Fig. 38 a, Picture of 1T/V₂O₃ film grown on sapphire substrate with different growth regions. Region A represents the V₂O₃ film without 1T sample. Region B represents the V₂O₃ film with isolated 1T flakes. Region C represents the V₂O₃ film with WS₂ flakes. Region D represents the very downstream area, which was not able to grow V₂O₃ film and WS₂ sample on the surface; **b**, Stitching optical micrograph of the interface between region A and region B. Please note that region B and region C are the main products, stitching optical micrograph was carried out for the clear observation of the morphology (see Fig. S39). Scale bar: b, 300 μm.

Supplementary Fig. 39 Stitching optical micrograph of as-grown 1T-WS₂ monolayers on the surface of V₂O₃/sapphire. Please note that the stitching area was selected from the interface between region B and Region C in the Fig. S38a. Scale bar: 500 μm.

Supplementary Fig. 40 a-d, STEM images of V SACs@1T-WS₂ merged film grown in region C. Single V atoms are marked by the white dashed circles. The STEM images were randomly taken from four different areas; **e**, Intensity sequence profiles of W-S-S-V-S-S-W (orange dashed arrow indicated by Figure S40a-d). Scale bars: 1 nm.

Supplementary Fig. 41 Raman spectrum of 1T merged film/V₂O₃ in growth region C.

Supplementary Fig. 42 a, Low-resolution cross-section HAADF-STEM image of V SACs@1T-WS₂ merged film on the surface of V₂O₃ film/sapphire substrate; **b**, High-resolution HAADF-STEM image of V SACs@1T-WS₂ monolayer film on the V₂O₃ surface; **c**, High-resolution HAADF-STEM image of the interface between V₂O₃ and sapphire substrate. Scale bars: a, 10 nm; b-c, 0.5 nm.

Accordingly, the description of characterizations of 1T merged film has been added in the revised supplementary manuscript to address this comment, which has been highlighted in Figure S42 as follows:

“The characterizations of V SACs@1T-WS₂ merged film in growth region C were shown in Supplementary Fig. 40-42.The monolayer thickness was confirmed by cross-sectional STEM images in Fig. S42.”

References

- 1 Elías, A. L. *et al.* Controlled Synthesis and Transfer of Large-Area WS₂ Sheets: From Single Layer to Few Layers. *ACS Nano* 7, 5235-5242 (2013).
- 2 Jeong, H. Y. *et al.* Heterogeneous Defect Domains in Single-Crystalline Hexagonal WS₂. *Adv. Mater.* 29, 1605043 (2017).
- 3 Gao, D. *et al.* Ferromagnetism in ultrathin VS₂ nanosheets. *J. Mater. Chem. C* 1, 5909-5916 (2013).
- 4 Green, C. L. & Kucernak, A. Determination of the Platinum and Ruthenium Surface Areas in Platinum-Ruthenium Alloy Electrocatalysts by Underpotential Deposition of Copper. I. Unsupported Catalysts. *J. Phys. Chem. B* 106, 1036-1047 (2002).
- 5 Wang, Y., Sofer, Z., Luxa, J. & Pumera, M. Lithium Exfoliated Vanadium Dichalcogenides (VS₂, VSe₂, VTe₂) Exhibit Dramatically Different Properties from Their Bulk Counterparts. *Adv. Mater. Interfaces* 3, 1600433 (2016).
- 6 Zhou, J. *et al.* Hierarchical VS₂ Nanosheet Assemblies: A Universal Host Material for the Reversible Storage of Alkali Metal Ions. *Adv. Mater.* 29, 1702061 (2017).
- 7 Yu, S. H. *et al.* In Situ Hybridizing MoS₂ Microflowers on VS₂ Microflakes in a One-Pot CVD Process for Electrolytic Hydrogen Evolution Reaction. *ACS Appl. Energy Mater.* 2, 5799-5808 (2019).
- 8 Silversmit, G. *et al.* A comparative XPS and UPS study of VO_x layers on mineral TiO₂ (001)-anatase supports. *Surf. Interface Anal.* 38, 1257-1265 (2006).
- 9 Canton-Vitoria, R. *et al.* Ping-Pong Energy Transfer in Covalently Linked Porphyrin-MoS₂ Architectures. *Angew. Chem. Int. Ed.* 59, 3976-3981, doi:10.1002/anie.201914494 (2020).
- 10 Zhao, X., Ma, X., Sun, J., Li, D. & Yang, X. Enhanced catalytic activities of surfactant-assisted exfoliated WS₂ nanodots for hydrogen evolution. *ACS nano* 10, 2159-2166 (2016).
- 11 Liu, L. *et al.* Phase-selective synthesis of 1T' MoS₂ monolayers and heterophase bilayers. *Nat. Mater.* 17, 1108-1114 (2018).
- 12 Qi, K. *et al.* Single-atom cobalt array bound to distorted 1T MoS₂ with ensemble effect for hydrogen evolution catalysis. *Nat. Commun.* 10, 5231 (2019).
- 13 Voiry, D. *et al.* Enhanced catalytic activity in strained chemically exfoliated WS₂ nanosheets for hydrogen evolution. *Nat. Mater.* 12, 850-855 (2013).
- 14 Wan, J. *et al.* In Situ Phosphatizing of Triphenylphosphine Encapsulated within Metal–Organic Frameworks to Design Atomic Co1–P₁N₃ Interfacial Structure for Promoting Catalytic Performance. *J. Am. Chem. Soc.* 142, 8431-8439 (2020).
- 15 Zhang, H., Yu, L., Chen, T., Zhou, W. & Lou, X. W. Surface Modulation of Hierarchical MoS₂ Nanosheets by Ni Single Atoms for Enhanced Electrocatalytic Hydrogen Evolution. *Adv. Funct. Mater.* 28, 1807086 (2018).
- 16 Anjum, M. A. R., Jeong, H. Y., Lee, M. H., Shin, H. S. & Lee, J. S. Efficient Hydrogen Evolution Reaction Catalysis in Alkaline Media by All-in-One MoS₂ with Multifunctional Active Sites. *Adv. Mater.* 30, 1707105 (2018).
- 17 Attanayake, N. H. *et al.* Effect of Intercalated Metals on the Electrocatalytic Activity of 1T-MoS₂ for the Hydrogen Evolution Reaction. *ACS Energy Lett.* 3, 7-13 (2018).
- 18 McCrory, C. C. L. *et al.* Benchmarking Hydrogen Evolving Reaction and Oxygen Evolving Reaction Electrocatalysts for Solar Water Splitting Devices. *J. Am. Chem. Soc.* 137,

- 4347-4357 (2015).
- 19 Wang, S. *et al.* Ultrastable In-Plane 1T–2H MoS₂ Heterostructures for Enhanced Hydrogen Evolution Reaction. *Adv. Energy Mater.* 8, 1801345 (2018).
- 20 Liu, Y. *et al.* Corrosion engineering towards efficient oxygen evolution electrodes with stable catalytic activity for over 6000 hours. *Nat. Commun.* 9, 2609 (2018).
- 21 McCrory, C. C. L., Jung, S., Peters, J. C. & Jaramillo, T. F. Benchmarking Heterogeneous Electrocatalysts for the Oxygen Evolution Reaction. *J. Am. Chem. Soc.* 135, 16977-16987 (2013).
- 22 Sievers, G. W. *et al.* Self-supported Pt–CoO networks combining high specific activity with high surface area for oxygen reduction. *Nat. Mater.*, doi:10.1038/s41563-020-0775-8 (2020).
- 23 Voiry, D. *et al.* Best Practices for Reporting Electrocatalytic Performance of Nanomaterials. *ACS Nano* 12, 9635-9638 (2018).
- 24 Yang, H. B. *et al.* Atomically dispersed Ni(i) as the active site for electrochemical CO₂ reduction. *Nat. Energy* 3, 140-147 (2018).
- 25 Chen, W. *et al.* Single Tungsten Atoms Supported on MOF-Derived N-Doped Carbon for Robust Electrochemical Hydrogen Evolution. *Adv. Mater.* 30, 1800396 (2018).
- 26 Cao, L. *et al.* Identification of single-atom active sites in carbon-based cobalt catalysts during electrocatalytic hydrogen evolution. *Nat. Catal.* 2, 134-141 (2019).
- 27 Deng, J. *et al.* Triggering the electrocatalytic hydrogen evolution activity of the inert two-dimensional MoS₂ surface via single-atom metal doping. *Energy Environ. Sci.* 8, 1594-1601.
- 28 Ji, Q. *et al.* Unravelling Orientation Distribution and Merging Behavior of Monolayer MoS₂ Domains on Sapphire. *Nano Lett.* 15, 198-205 (2015).
- 29 van der Zande, A. M. *et al.* Grains and grain boundaries in highly crystalline monolayer molybdenum disulphide. *Nat. Mater.* 12, 554-561 (2013).
- 30 Yu, H. *et al.* Wafer-Scale Growth and Transfer of Highly-Oriented Monolayer MoS₂ Continuous Films. *ACS Nano* 11, 12001-12007 (2017).
- 31 Zhang, Q. *et al.* Reliable Synthesis of Large-Area Monolayer WS₂ Single Crystals, Films, and Heterostructures with Extraordinary Photoluminescence Induced by Water Intercalation. *Adv. Optical Mater.* 6, 1701347 (2018).
- 32 Chow, P. K. *et al.* Wetting of Mono and Few-Layered WS₂ and MoS₂ Films Supported on Si/SiO₂ Substrates. *ACS Nano* 9, 3023-3031 (2015).
- 33 Perea-López, N. *et al.* Photosensor Device Based on Few-Layered WS₂ Films. *Adv. Funct. Mater.* 23, 5511-5517, doi:10.1002/adfm.201300760 (2013).

REVIEWER COMMENTS

Reviewer #3 (Remarks to the Author):

Overall, the V-XP spectra for the transferred films are too weak. In addition, for the STEM simulations, the authors did not compare the V atom with the S vacancy having one W atom and a S atom, so it is very important to revise the line profile.

My comments are as follows:

1. For Fig.S6, could the authors overlay the experimental and simulation line profile together, and normalize the peak intensity based on the intensity of the W peak. In this way, one could clearly see the differences. For example, the experimental line profile indicates that the V peak intensity is about the same as the 2nd S next to it. However, in the simulation, V is slightly higher than that. In addition, for the W vacancies, did the authors removed W+2S? What about the vacancy with only W+S?
2. Even by increasing the acquisition time to 120s, the V signal seems too weak if it is 4at%. It is still not quite convincing that V atoms are indeed present in the films.

Reviewer #3 (Remarks to the Author):

Overall, the V-XP spectra for the transferred films are too weak. In addition, for the STEM simulations, the authors did not compare the V atom with the S vacancy having one W atom and a S atom, so it is very important to revise the line profile.

Response. We appreciate the reviewer very much for his/her comments. His/Her concerns of weak V 2p signal in the transferred film and STEM simulations are addressed as follows.

Comment 1. *For Fig.S6, could the authors overlay the experimental and simulation line profile together, and normalize the peak intensity based on the intensity of the W peak. In this way, one could clearly see the differences. For example, the experimental line profile indicates that the V peak intensity is about the same as the 2nd S next to it. However, in the simulation, V is slightly higher than that. In addition, for the W vacancies, did the authors removed W+2S? What about the vacancy with only W+S?*

Response 1. We appreciate the reviewer very much for these very good suggestions. **Firstly**, we have overlaid the experimental and simulated line intensity profiles together, as shown in Figure S6c. From the result, the experimental profile showed a coincident W/V peak intensity ratio with the simulated one after normalizing the peak intensity based on the intensity of the W peak. Please note that an average experimental line intensity profile was obtained from 27 different line intensity profiles with W-S-S-V-S-S-W sequence in Figure R1a (from Figure 2c in the revised manuscript). **Secondly**, the reviewer mentioned that different peak intensity ratios of V and S (near to V) were observed in the previously experimental and simulated profiles. It's worth emphasizing that the peak intensity (even for the same element) was slightly different due to the limitation of detection accuracy, which has been observed in many published literatures.^{1,2} For example, the W peak intensity and V peak intensity profile in the V-doped 2H-WSe₂ showed slightly different intensity in both W-Se-V-Se-W-Se sequence and W-W-V-V-W-W sequence (Figure S3, Adv. Mater. 2020, 2003607)². **We have cited this paper in the revised manuscript**

(please see reference 36). In the case of present V VSACs@1T-WS₂ sample, the peak intensity ratio of V or S was slightly different if choosing different line sequence in our sample. Hence, to obtain a convincing experimental intensity profile, we obtain an average experimental line intensity profile from 27 different line intensity profiles with W-S-S-V-S-S-W sequences (Figure R1a). And the final V peak intensity is slightly higher than the S peak intensity, which is generally consistent with the simulation ones. **Thirdly**, we appreciate your suggestions of “*In Figure 2c, could the authors provide the intensity profile of V SACs@1T-WS₂ with V atoms, and compare it with the simulation? Figure 2f only shows the line profile with W and S atoms. The profiles with V, W, and S can help distinguish the V@W sites to vacancy@W sites*” in the last round questions. According to your suggestions, we have made simulated STEM images with both V SACs@1T-WS₂ and W_{vac}@1T-WS₂ to address this comment. As the much lower atomic number of V than W, V atoms in the samples were not visible as well as the simulated STEM image in Figure S6b. Hence, it was very necessary to compare the experimental V SACs@1T-WS₂ with the simulated W_{vac}@1T-WS₂. So, we compared the experimental line intensity profile of V SACs@1T-WS₂ with simulated line intensity profiles of V SACs@1T-WS₂ (Figure S6b) and W_{vac}@1T-WS₂ (Figure S6c) from W-S-S-V-S-S-W (Figure S6) and W-W-V-W-W sequences (Figure S7), respectively. The experimental line intensity profile of V SACs@1T-WS₂ in Figure S7 was obtained from an average 28 different line intensity profiles with W-W-V-W-W sequences (Figure R1b). **Fourthly**, the reviewer was wondering the coordination of W_{vac} in the previous simulated line intensity profiles of W_{vac}@1T-WS₂. In that simulation, we only considered W_{vac} defect with the removal of W atom, as shown in Figure S6d. Additionally, the W_{vac} defect with the removal of W atom and 2S vacancy was also shown in Figure S6e. The corresponding intensity profiles were shown in orange dots and pink dots in Figure S6h, respectively. **Finally**, according to your suggestion, we have also considered only one S vacancy defect with one W and S atom and compared it with 1T-WS₂ without any vacancy defect, as shown in Figure S6f and Figure S6g, respectively. The corresponding intensity profiles were shown in olive dots and blue

dots in Figure S6h, respectively.

Figure R1 a, Experimental STEM image of V SACs@1TWS₂. The 27 different red dot lines represent the sequences of W-S-S-V-S-S-W; **b**, Experimental STEM image of V SACs@1TWS₂. The 28 different yellow dot lines represent the sequences of W-W-V-W-W.

The above-mentioned Supplementary Fig. 6 and Supplementary Fig. 7 have been changed in the revised supporting information as follows:

Supplementary Fig. 6 a-b, Experimental (a) and simulated (b) STEM images of V SACs@1TWS₂, respectively; **c**, Corresponding experimental (purple) and simulated (black dots) intensity sequence profiles of W-S-S-V-S-S-W (red dashed arrow) indicated by the STEM images. The white dashed circles represent the V atoms; **d-e**, Simulated STEM images of W_{vac}@1T-WS₂ with no S vacancy (d) and 2S vacancies (e). The purple dashed circles and yellow dashed circles represent the W_{vac} and S_{vac}, respectively; **f-g**, Simulated STEM images of 1T-WS₂ with 1S vacancy (f) and no S vacancy (g); **h**, Corresponding experimental (purple) intensity sequence profile of W-S-S-V-S-S-W (red dashed arrow) indicated by Figure S6a, simulated (orange dots) intensity sequence profiles of W-S-S-W_{vac}-S-S-W (red dashed arrow) indicated by the Figure S6d, simulated (pink dots) intensity sequence profiles of W-S-S_{vac}-W_{vac}-S_{vac}-S-S-W (red dashed arrow) indicated by the Figure S6e, simulated (olive dots) intensity sequence profiles of W-S-S-W-S-S_{vac}-W (red dashed arrow) indicated by the Figure S6f and simulated (blue dots) intensity sequence profiles of W-S-S-W-S-S-W (red dashed arrow) indicated by the Figure S6g. The blue spheres, red spheres and yellow spheres represent the W atoms, V atoms and S atoms, respectively.

Supplementary Fig. 7 a-b, Experimental (a) and simulated (b) STEM images of V SACs@1T-WS₂, respectively. The white dashed circles represent the V atoms; **c**, Simulated STEM image of 1T-WS₂ with vacancies at W sites (W_{vac}@1T-WS₂). The purple dashed circles represent the vacancies at W sites; **d**, Corresponding intensity sequence profiles of W-W-V-W-W or W-W-W_{vac}-W-W indicated by the STEM images (yellow dashed arrow). The blue spheres, and red spheres in Figure S7a-7c represent the W atoms and V atoms, respectively.

Comment 2. *Even by increasing the acquisition time to 120s, the V signal seems too weak if it is 4at%. It is still not quite convincing that V atoms are indeed present in the films.*

Response 2. We appreciate the reviewer very much for this very important concern. The weak V 2p signal was probably caused by the single layer thickness of 1T sample, as a result, the detected signals were mainly from HOPG substrate. To address this comment, we have made efforts to highlight the importance of enough 1T sample to obtain the enhanced V 2p signals. **Firstly**, to further verify the V signal in the 1T sample, we transferred the 1T samples on the same HOPG area for 15 times to acquire the XPS spectra, as the XPS technique was able to detect the surface of sample in 10

nm depth. As shown in Figure S12c, an enhanced V 2p signal was observed, matching well with the V 2p signal in Figure S11c or other reported V 2p signals in VS₂.^{3,4} **Secondly**, we also transferred the 1T samples that were obtained at 840 °C to confirm the V signals, as shown in Figure R2. As demonstrated in Figure S37a, the 1T samples obtained at 840 °C showed a much higher V concentration, so the V signals could be easily detected. **Finally**, except for the XPS demonstration of V signals in 1T sample, EELS spectrum in Figure 2e and line intensity profiles analyses in Figure S6-S7 were all powerful evidence to demonstrate the presence of V elements.

Accordingly, the demonstration of enhanced V 2p signals has been added in the revised supporting manuscript to address this comment, which has been highlighted on Figure S12 as follows:

“As the XPS technique was able to detect the surface of sample in 10 nm depth. To further verify the V signal in the 1T sample, we transferred the 1T samples on the same HOPG area for 15 times to acquire the XPS spectra. As shown in Figure S12c, an enhanced V 2p signal was observed, matching well with the V 2p signal in Figure S11c.

”

“Please note that However, the molar ratio of W/V was ~95:5 (Figure S11) or 16:1 (Figure S12) from the XPS analyses, which was close to the result obtained by the STEM image (~4 at%), further confirming the absence of V-based contaminations in the transferred 1T sample. In addition, only C-O bond at 532.4 eV (Figure S11d and S12d) or C=O bond at 533.9 eV (Figure S12d) were obtained and no V-O bond from vanadium oxidation (530.0 eV) was observed from the O 1s peak, further excluding the possibility of V₂O₃ or VO₂ in the transferred 1T sample.”

The above-mentioned Supplementary Fig. 12 has been added in the revised supporting information as follows:

Supplementary Fig. 12 a-d, High-resolution XPS spectra of W 4f (a), S 2p (b), V 2p (c) and O 1s (d) core levels of V SACs@1T-WS₂ transferred on HOPG substrate for 15 times. The fitting blue and pink curves represent the contributions of 1T and 2H phases in Fig. S12 a-b, respectively. The molar ratio of W/V obtained by the XPS analysis was ~16:1.

Figure R2 High-resolution XPS spectrum of V 2p core level of V SACs@1T-WS₂ obtained at 840 °C transferred on HOPG substrate for 15 times.

Reference

- Zhang, F. *et al.* *Monolayer Vanadium-doped Tungsten Disulfide: A Room-Temperature Dilute Magnetic Semiconductor.* (2020).

<https://arxiv.org/abs/2005.01965>

- 2 Pham, Y. T. H. *et al.* Tunable Ferromagnetism and Thermally Induced Spin Flip in Vanadium-Doped Tungsten Diselenide Monolayers at Room Temperature. *Adv. Mater.*, doi:<https://doi.org/10.1002/adma.202003607> (2020).
- 3 Zhou, J. *et al.* Hierarchical VS₂ Nanosheet Assemblies: A Universal Host Material for the Reversible Storage of Alkali Metal Ions. *Adv. Mater.* **29**, 1702061 (2017).
- 4 Liang, H. *et al.* Solution Growth of Vertical VS₂ Nanoplate Arrays for Electrocatalytic Hydrogen Evolution. *Chem. Mater.* **28**, 5587-5591 (2016).

REVIEWERS' COMMENTS

Reviewer #3 (Remarks to the Author):

The authors have addressed the comments and I am happy to recommend publication.